EMBO
Molecular Medicine

# An FGFR3/MYC positive feedback loop provides new opportunities for targeted therapies in bladder cancers

Mélanie Mahe[1,2,†], Florent Dufour[1,2,†], Hélène Neyret-Kahn[1,2], Aura Moreno-Vega[1,2], Claire Beraud[3], Mingjun Shi[1,2], Imene Hamaidi[4], Virginia Sanchez-Quiles[1,2], Clementine Krucker[1,2], Marion Dorland-Galliot[1,2], Elodie Chapeaublanc[1,2], Remy Nicolle[1,2], Hervé Lang[4], Celio Pouponnot[5,6,7], Thierry Massfelder[8], François Radvanyi[1,2] & Isabelle Bernard-Pierrot[1,2,*] 🆔

## Abstract

FGFR3 alterations (mutations or translocation) are among the most frequent genetic events in bladder carcinoma. They lead to an aberrant activation of FGFR3 signaling, conferring an oncogenic dependence, which we studied here. We discovered a positive feedback loop, in which the activation of p38 and AKT downstream from the altered FGFR3 upregulates *MYC* mRNA levels and stabilizes MYC protein, respectively, leading to the accumulation of MYC, which directly upregulates *FGFR3* expression by binding to active enhancers upstream from *FGFR3*. Disruption of this FGFR3/MYC loop in bladder cancer cell lines by treatment with FGFR3, p38, AKT, or BET bromodomain inhibitors (JQ1) preventing *MYC* transcription decreased cell viability *in vitro* and tumor growth *in vivo*. A relevance of this loop to human bladder tumors was supported by the positive correlation between *FGFR3* and *MYC* levels in tumors bearing *FGFR3* mutations, and the decrease in FGFR3 and MYC levels following anti-FGFR treatment in a PDX model bearing an *FGFR3* mutation. These findings open up new possibilities for the treatment of bladder tumors displaying aberrant FGFR3 activation.

**Keywords** BET inhibitors; bladder cancer; FGFR3; MYC; p38
**Subject Categories** Cancer; Urogenital System

## Introduction

Bladder cancer is the ninth most common cancer worldwide, with approximately 430,000 new cases diagnosed in 2012 and 165,000 deaths annually (Antoni *et al*, 2017). Non-muscle-invasive carcinomas (NMIBCs) account for 70% of cases at first diagnosis. These tumors often have a favorable prognosis following transurethral resection with or without intravesical chemotherapy or immunotherapy with Bacillus Calmette-Guérin (BCG). NMIBC often recurs (50–60% of cases) and sometimes progresses to a muscle-invasive tumor (5–40% progression, depending on clinical and pathological features). This high recurrence rate and the need for monitoring contribute to the economic burden of bladder cancer treatment. Muscle-invasive bladder carcinoma (MIBC) is a major clinical issue, because, even with radical cystectomy as the standard treatment, overall survival at 5 years is only about 50%, and the combination of this treatment with neoadjuvant and/or adjuvant chemotherapy increases overall survival only moderately. No major improvement in survival has been achieved over the last 20 years (Witjes *et al*, 2013). A clinical response to immune checkpoint inhibitors has recently been reported, but only a subset of patients respond to such treatment, and it remains unclear how to identify these patients (Powles *et al*, 2014; Bajorin *et al*, 2015; Bellmunt *et al*, 2017a,b; Davarpanah *et al*, 2017). Some targeted therapies have also yielded promising efficacy results. This is the case, for example, for mTOR inhibitors for patients with TSC1 mutations, anti-HER2 treatments for HER2-amplified MIBC, and anti-FGFR therapies for MIBC with activating FGFR mutations or translocations (Abbosh *et al*, 2015; Rouanne *et al*, 2016). The definition of

1 Institut Curie, CNRS, UMR144, Equipe Labellisée Ligue contre le Cancer, PSL Research University, Paris, France
2 CNRS, UMR144, Sorbonne Universités, UPMC Université Paris 06, Paris, France
3 UROLEAD SAS, School of Medicine, Strasbourg, France
4 Department of Urology, Nouvel Hôpital Civil, Hôpitaux Universitaires de Strasbourg, Strasbourg, France
5 Institut Curie, Orsay, France
6 CNRS UMR3347, Centre Universitaire, Orsay, France
7 INSERM U1021, Centre Universitaire, Orsay, France
8 INSERM UMR_S1113, Section of Cell Signalization and Communication in Kidney and Prostate Cancer, School of Medicine, Fédération de Médecine Translationnelle de Strasbourg (FMTS), INSERM and University of Strasbourg, Strasbourg, France
*Corresponding author. Tel: +33 1 42 34 63 40; Fax: +33 1 42 34 63 49; E-mail: isabelle.bernard-pierrot@curie.fr
†These authors contributed equally to this work

therapeutic strategies to improve treatment outcomes remains of the utmost importance.

FGFR3 (fibroblast growth factor receptor) belongs to a family of structurally related tyrosine kinase receptors (FGFR1-4). These receptors regulate various physiological processes, including proliferation, differentiation, migration, and apoptosis. There has been considerable interest in the FGFR family (FGFR1-4), as these receptors are frequently involved, through various mechanisms, in genetic disorders and cancer, leading to their identification as possible targets for treatment (Haugsten *et al*, 2010). FGFR3 is frequently altered through activating mutations and translocations generating FGFR3-gene fusions (Billerey *et al*, 2001; Tcga, 2014). Mutations are, by far, the most frequent alterations of *FGFR3*, occurring in almost 50% of bladder tumors (70% of NMIBCs and 15–20% of MIBCs). The two most frequent mutations are the S249C and Y375C mutations, which affect the extracellular domain of the receptor. *FGFR3* translocations leading to the production of FGFR3-TACC3 and FGFR3-BAIAP2L1 fusion proteins were recently identified in 3% of MIBCs (Tcga, 2014). These alterations are thought to be "oncogenic drivers", because the expression of an altered FGFR3 induces cell transformation (Bernard-Pierrot *et al*, 2006; Williams *et al*, 2013; Wu *et al*, 2013; Nakanishi *et al*, 2015). Furthermore, several preclinical studies in cell lines and xenograft models of bladder cancer have shown that FGFR3 alterations confer sensitivity to FGFR inhibitors, which have anti-proliferative and pro-apoptotic effects (Bernard-Pierrot *et al*, 2006; Wu *et al*, 2013; Nakanishi *et al*, 2015). Together, these findings highlight the critical role of FGFR3 in bladder tumor carcinogenesis, raising the possibility of developing anti-FGFR3 therapies for both NMIBC and MIBC (Chae *et al*, 2017). Promising results were recently reported for four out of the five patients with FGFR3-mutated bladder cancers enrolled in a phase I clinical trial of the pan-FGFR kinase inhibitor BGJ398 (Nogova *et al*, 2017). However, based on observations for other targeted therapies (EGFR, BRAF, KIT) for various cancers, including colon and lung cancers, melanoma, and gastrointestinal tumors, FGFR3-targeted therapies will probably turn out to be limited by multiple mechanisms of intrinsic and acquired resistance, such as ERBB2/3 or EGFR activation (Flaherty *et al*, 2012; Herrera-Abreu *et al*, 2013; Niederst & Engelman, 2013; Wang *et al*, 2015). The signaling pathway activated by mutated FGFR3 and FGFR3-fusion proteins is not well characterized, particularly for bladder cancer. Improvements in our understanding of the molecular mechanisms underlying the oncogenic activity of activated FGFR3 in bladder tumors may facilitate the identification of new drug targets that could be acted on together with FGFR3, to increase the efficacy of anti-FGFR3 therapies and/or to prevent potential drug resistance. Such strategies, based on the simultaneous inhibition of two or more targets in a single pathway, have already been explored for many specific pairs of agents, in both clinical and preclinical studies (Flaherty *et al*, 2012; Li *et al*, 2014; Ran *et al*, 2015). In this study, we aimed to characterize the aberrantly activated FGFR3 signaling pathways involved in bladder cancer cell growth/transformation. We studied genes regulated by constitutively activated FGFR3 in two bladder tumor-derived cell lines, MGH-U3 and RT112, harboring an *FGFR3* mutation (Y375C) and a fusion gene (FGFR3-TACC3), respectively. We identified MYC as a key transcription factor that is overexpressed and activated in response to FGFR3 activity, and critical for FGFR3-induced cell proliferation. We showed here that

*FGFR3* is a direct target gene of MYC, which binds to active enhancers located upstream from *FGFR3*, establishing an FGFR3/MYC positive feedback loop. This loop may be relevant in human tumors, because *MYC* and *FGFR3* expression levels were found to be positively correlated in tumors bearing *FGFR3* mutations in two independent transcriptomic datasets ($n = 63$ and $n = 271$), and because FGFR3 inhibition in a patient-derived tumor xenograft (PDX) model harboring an FGFR3-S249C mutation decreased the levels of both MYC and FGFR3. We found that *MYC* mRNA levels and protein stability were dependent on p38 and AKT activation, respectively, downstream from FGFR3 activation. Finally, we showed, in xenograft models, that FGFR3 activation conferred sensitivity to FGFR3 and p38 inhibitors and to a BET bromodomain inhibitor (JQ1) preventing *MYC* transcription. These findings therefore suggest new treatment options for bladder cancers in which FGFR3 is aberrantly activated.

# Results

## MYC is a key master regulator of proliferation in the aberrantly activated FGFR3 pathway

We investigated the molecular mechanisms underlying the oncogenic activity of aberrantly activated FGFR3 in bladder carcinomas, by studying the MGH-U3 and RT112 cell lines. These cell lines were derived from human bladder tumors, and they endogenously express a mutated activated form of FGFR3 (FGFR3-Y375C, the second most frequent mutation in bladder tumors) and the FGFR3-TACC3 fusion protein (the most frequent FGFR3 fusion protein in bladder tumors), respectively. The growth and transformation of these cell lines are dependent on FGFR3 activity (Bernard-Pierrot *et al*, 2006; Williams *et al*, 2013; Wu *et al*, 2013). We conducted a gene expression analysis with Affymetrix DNA arrays, in these cell lines, with and without *FGFR3* siRNA treatment. We identified 741 and 3,124 genes displaying significant differential expression after *FGFR3* depletion in MGH-U3 and RT112 cells, respectively (adjusted $P$-values < 0.05, |log2(FC)| > 0.5; Dataset EV1). An analysis of these two lists of FGFR3-regulated genes using the upstream regulator function of Ingenuity Pathway Analysis (IPA) software identified upstream regulators activated and inhibited by FGFR3 (Fig 1A, left panel). The top 10 transcriptional regulators with activity modulated by FGFR3 were common to the two cell lines and are listed in the right panel in Fig 1A. The transcription factor predicted to be the most strongly inhibited here after *FGFR3* depletion, in both cell lines, was the proto-oncogene MYC, for which mRNA levels were downregulated. This downregulation of *MYC* mRNA levels after *FGFR3* knockdown with siRNA was further confirmed by reverse transcription–quantitative polymerase chain reaction (RT-qPCR) (30–70% decrease, depending on the cell line used; Fig 1B). Consistent with these results suggesting that *MYC* mRNA levels are modulated by constitutively activated FGFR3, an analysis of previously described transcriptomic data for our CIT-series ("*Carte d'Identité des Tumeurs*"; tumor identity card) of bladder tumors revealed a significant upregulation of *MYC* mRNA levels in tumors harboring an *FGFR3* mutation ($n = 63$) relative to normal urothelium samples ($n = 4$), whereas no such overexpression was observed for tumors expressing wild-type *FGFR3* ($n = 122$; Fig 1C). Moreover, *MYC*

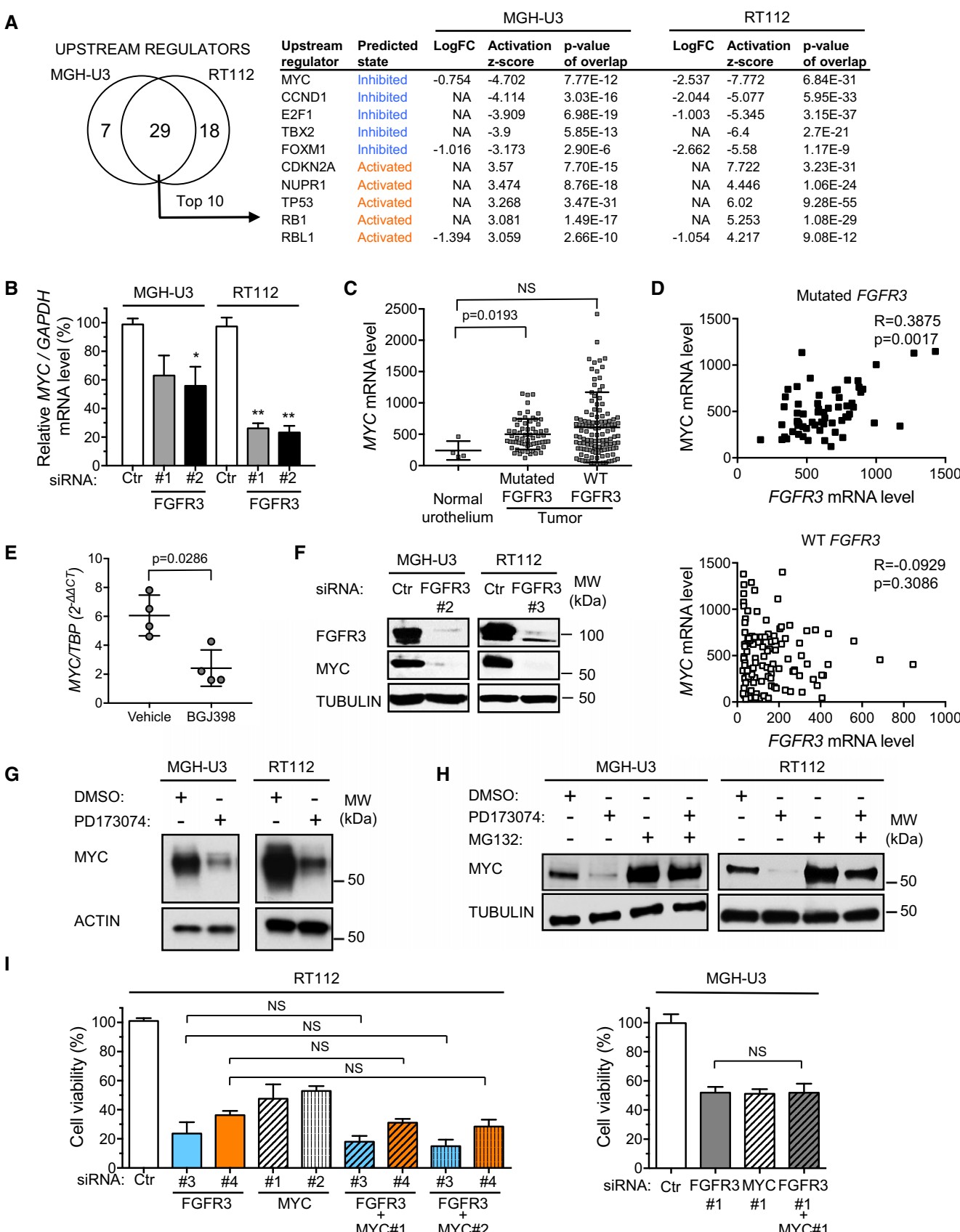

**Figure 1.**

**Figure 1.  MYC is a key upstream regulator activated by FGFR3 that is required for FGFR3-induced bladder cancer cell growth.**

A   Venn diagram showing the number of upstream regulators (transcription factors) significantly predicted by Ingenuity Pathway Analysis to be involved in the regulation of gene expression observed after *FGFR3* knockdown in RT112 and MGH-U3 cells (left panel). List of the top 10 upstream regulators modulated by FGFR3 expression in both cell lines. The Log$_2$FC of the transcription factor itself is also indicated. NA indicates that the FC was beyond the threshold defining genes as differentially expressed after *FGFR3* depletion (see Materials and Methods).

B   Relative *MYC* mRNA levels in MGH-U3 and RT112 cells transfected for 72 h with siRNAs targeting *FGFR3* or a control siRNA (Ctr). The results presented are the means of two independent experiments carried out in triplicate; the standard errors are indicated. The significance of differences was assessed in unpaired Student's *t*-tests, $*P < 0.05$; $**0.001 < P < 0.005$.

C   *MYC* mRNA levels in normal human urothelium ($n = 4$) and in the CIT cohort of human bladder tumors bearing *FGFR3* mutations ($n = 63$) or wild-type *FGFR3* ($n = 122$). The significance of differences was assessed in Mann–Whitney tests, and means and standard errors are represented.

D   *MYC* and *FGFR3* mRNA levels in human bladder tumors harboring either mutated *FGFR3* (upper panel) or wild-type *FGFR3* (lower panel). Spearman's coefficient and *P*-values are indicated for the correlations between *MYC* and *FGFR3* mRNA levels in each group.

E   *MYC* mRNA levels in a PDX model bearing a FGFR3-S249C mutation and treated daily, for 4 days, with 30 mg/kg BGJ398, a pan-FGFR inhibitor, or with vehicle ($n = 4$ mice per group). Means and standard errors are represented. The significance of differences was assessed in Mann–Whitney tests.

F   Western blot (72 h after transfection) comparing FGFR3 and MYC levels in MGH-U3 and RT112 cells transfected with a control siRNA (Ctr) or with siRNAs targeting *FGFR3*.

G   Western blot comparing MYC levels in MGH-U3 and RT112 cells, treated for 2 h with DMSO or the pan-FGFR inhibitor, PD173074 (500 nM).

H   Western blot comparing MYC levels in MGH-U3 and RT112 cells treated for 3 h with FGFR inhibitor (0.5 μM PD173074) or proteasome inhibitor (10 μM MG132), alone or in combination.

I   Cell viability assay comparing the impact of *MYC* and/or *FGFR3* downregulation on RT112 (left panel, CellTiter-Glo) and MGH-U3 (right panel, MTT assay) cell viability (72 h post-transfection). The results presented are the means of three independent experiments carried out in triplicate, error bars represent standard deviations. Tukey's multiple comparisons tests were performed to evaluate the significance of differences. The results of the statistical analysis are summarized in Dataset EV2.

Source data are available online for this figure.

expression was positively correlated with *FGFR3* expression in bladder tumors harboring a mutated *FGFR3* (Fig 1D, upper panel), whereas no such correlation was observed in tumors bearing wild-type *FGFR3* ($n = 122$; Fig 1D, lower panel). Similar results were also observed for another publicly available transcriptomic dataset for 416 bladder tumors (271 with *FGFR3* mutations) and eight normal samples (Hedegaard *et al*, 2016; Appendix Fig S1A and B), suggesting that mutated *FGFR3* may also regulate *MYC* expression in human bladder carcinomas. Support for this hypothesis was provided by the significant decrease in *MYC* mRNA levels induced by 4 days of anti-FGFR treatment in tumors from a PDX model (F659) bearing an FGFR3-S249C mutation (Fig 1E). As in cell lines, FGFR3-S249C expression conferred FGFR3 dependence on the PDX model, in which anti-FGFR treatment with BGJ398 decreased tumor growth by 60% after 29 days of administration (Appendix Fig S2).

MYC is a key regulator of proliferation and its deregulation can promote oncogenesis in various types of cancer (Dang, 2012). We therefore investigated the role of MYC as a master regulator of proliferation in bladder cell lines expressing aberrantly activated FGFR3. Western blot analysis further showed that *FGFR3* depletion resulted in the almost total loss of MYC from both MGH-U3 and RT112 cells (Fig 1F). The discrepancy between the decreases in *MYC* mRNA (Fig 1B) and protein levels (Fig 1F) suggested that the aberrant activation of FGFR3 regulated MYC not only at mRNA level, but also through stabilization of the protein. This hypothesis was also supported by the time course of MYC expression on Western blots after the inhibition of FGFR3 with PD173074. Indeed, MYC levels decreased rapidly, after 30 min of treatment, in MGH-U3 cells (Appendix Fig S3A), and expression was totally lost after 2 h of treatment, in both MGH-U3 and RT112 cells (Fig 1G and Appendix Fig S3A). MYC protein stability is, thus, tightly controlled by the proteasome. We therefore investigated the possible role of FGFR3 in this process, by treating MGH-U3 and RT112 cells with a pan-FGFR inhibitor (PD173074), either alone or in combination with a proteasome inhibitor (MG132; Fig 1H). Western blot analysis showed that the downregulation of MYC induced by the

inhibition of FGFR3 was abolished by MG132, in both cell lines. Overall, our results indicate that the inhibition of aberrantly activated FGFR3 decreases *MYC* mRNA levels and favors proteolysis of the MYC protein by the proteasome, thereby decreasing its transcriptional activity. We then investigated the possible contribution of MYC to the oncogenic activity of aberrantly activated FGFR3. We compared the effects on viability of depleting *FGFR3* and *MYC* alone or together, with siRNA, in RT112 and MGH-U3 cells (Fig 1I). *FGFR3* and *MYC* siRNAs efficiently knocked down the levels of the targeted proteins (Appendix Fig S3B). The depletion of either *MYC* or *FGFR3* resulted in significantly lower cell viability than for cells treated with the control siRNA (Fig 1I, right and left panels and Dataset EV2 for the *P*-values). No significant additive effect relative to FGFR3 depletion alone was observed in RT112 and MGH-U3 cells with a simultaneous knockdown of *FGFR3* and *MYC* expression, suggesting that MYC is a key downstream effector of the aberrantly activated FGFR3 pathway mediating cell proliferation.

**FGFR3 and MYC are involved in a positive feedback loop in which FGFR3 is a direct transcriptional target of MYC in bladder cancer cell lines with constitutively activated FGFR3**

Surprisingly, we observed that the treatment of MGH-U3 and RT112 cells with a *MYC* siRNA strongly decreased FGFR3 levels (Fig 2A). RT–qPCR showed that this loss of FGFR3 expression was due to a decrease in *FGFR3* mRNA levels after *MYC* knockdown (Fig 2B). We investigated whether *FGFR3* was a direct transcriptional target of MYC, by analyzing MYC occupancy of the *FGFR3* locus by chromatin immunoprecipitation and quantitative PCR (ChIP–qPCR). Using the publicly available ENCODE data for three different cancer cell lines, we designed primers binding to two potential enhancers, the promoter and an intragenic region of *FGFR3* (Appendix Fig S4A). According to ENCODE data, the enrichment of MYC and activation marks (H3K27ac) in the E1 and E2 enhancers is correlated with the level of *FGFR3* transcription (Appendix Fig S4A). We

checked, by ChIP–qPCR, that the selected FGFR3 promoter and enhancers did harbor the expected histone activation marks (H3K27ac and H3K4me3) in RT112 cells (Appendix Fig S4B). Finally, we showed that the two *FGFR3* enhancer regions tested were enriched in MYC, consistent with the direct regulation of *FGFR3* expression by MYC, at the transcriptional level (Fig 2C). This regulation of *FGFR3* by MYC seemed to be quite specific to bladder cancer, because MYC binding to the *FGFR3* enhancers or promoter was rarely observed in a publicly available dataset encompassing 118 MYC chromatin immunoprecipitation and sequencing (ChIP-Seq) in different tissues (Appendix Fig S5A). Binding was observed in two known FGFR3-dependent cell lines, MCF7 and HepG2 (Qiu *et al*, 2005; Tomlinson *et al*, 2012), in some blood-derived cell lines and in one lung cancer-derived cell line. MYC activation did not seem to be sufficient to induce FGFR3 regulation. Indeed, MYC ChIP-Seq data acquired for two inducible models of MYC overexpression/activation (LNCaP and U2OS cells; Walz *et al*, 2014; Barfeld *et al*, 2017) showed no MYC enrichment on the *FGFR3* enhancers or promoter after MYC activation (Appendix Fig S5B and C). Our data therefore identify MYC as a master regulator of proliferation activated downstream from FGFR3 (Fig 1) and as a positive regulator of FGFR3 expression in bladder cancer lines (Fig 2A–C). Consistent with this FGFR3/MYC positive feedback loop, we also observed that the treatment of RT112 and MGH-U3 cells with a pan-FGFR kinase inhibitor abolished both MYC and FGFR3 expression (Fig 2D). This result was confirmed in two other cell lines expressing constitutively activated FGFR3: UM-UC-14 (FGFR3-S249C) and RT4 (FGFR-TACC3 breakpoint exon 18 FGFR3–exon 4 TACC3, whereas FGFR3-TACC3 breakpoint exon 18 FGFR3–exon 11 TACC3 is expressed in RT112; Williams *et al*, 2013; Earl *et al*, 2015; Fig 2D). These four cell lines express low levels of *FGFR1, FGFR2,* and *FGFR4,* as assessed with an Affymetrix U133plus2 array, suggesting that the observed effect was mostly due to FGFR3 inhibition (data not shown). However, treatment had no effect on MYC and FGFR3 expression in UM-UC-5 cells, which express wild-type FGFR3 (Fig 2D). These results suggest that the FGFR3/MYC positive feedback loop is a general mechanism, regardless of the type of FGFR3 alteration, but that it is dependent on activated FGFR3. Using RT112 and MGH-U3 xenograft models treated for 9 days with a pan-FGFR inhibitor, PD173074, which delayed tumor growth (Appendix Fig S6A), we also showed *in vivo* that FGFR3 and MYC were involved in a positive feedback circuit inducing bladder tumor growth. Indeed, immunoblot analysis revealed that FGFR3 inhibition resulted in lower levels of both MYC and FGFR3 in the xenografts (Fig 2E). Finally, we made use of our PDX model (F659) harboring an FGFR3-S249C mutation to demonstrate that this FGFR3/MYC loop was relevant to human tumors. Indeed, the treatment of tumor-bearing mice for 4 days with another pan-FGFR inhibitor, BGJ398, which inhibited PDX tumor growth (Appendix Fig S6B), decreased both MYC and FGFR3 levels in the tumors (Fig 2F).

## MYC accumulation induced by aberrantly activated FGFR in bladder tumors depends on p38 and AKT activation

Given the importance of the FGFR3/MYC loop in all our tested models, including the PDX model, we characterized the underlying mechanisms. We investigated the signals downstream from FGFR

responsible for the observed higher levels of *MYC* mRNA and greater MYC protein stability in bladder cancer cells harboring FGFR3 mutations (Fig 1).

We first used transformed NIH-3T3 cells expressing FGFR3-S249C established in a previous study (Bernard-Pierrot *et al*, 2006) to confirm that mutated FGFR3 expression induced an upregulation of *MYC* mRNA levels (Appendix Fig S7A). We investigated the activation of three pathways known to be activated by tyrosine kinase receptors and, in particular, FGFRs (p38, AKT, ERK1/2; Powers *et al*, 2000; Appendix Fig S7B), and evaluated their role in the cell transformation induced by mutated FGFR3 (Appendix Fig S7C). We found that the activation of p38 and AKT mediated cell transformation downstream from the mutated FGFR3 whereas ERK1/2 activation was less crucial for FGFR3 activity. It has been established that p38 can induce the stabilization of *MYC* mRNA or the upregulation of MYC protein levels through an increase in transcription (Chen *et al*, 2005) whereas AKT can induce the stabilization of MYC protein (Tsai *et al*, 2012). We thus investigated the involvement of these two pathways in our urothelial models, MGH-U3 and RT112 cells. We showed that p38 and AKT were constitutively activated in both cell lines. This activation was dependent on FGFR3 expression, because it was abolished by *FGFR3* knockdown (Fig 3A). We then explored the role of p38 in the FGFR3-induced upregulation of *MYC* mRNA levels, using a p38 siRNA targeting *MAPK14* (p38α), the predominant isoform in MGH-U3 and RT112 cells, as shown by Affymetrix U133 plus 2.0 DNA chip analyses (data not shown). Immunoblot analysis showed that the efficient depletion of p38 resulted in the loss of about 50% of MYC in both MGH-U3 and RT112 cells, whereas MYC loss was total following *FGFR3* depletion (Fig 3B). This decrease in MYC levels is consistent with the decrease in *MYC* mRNA levels observed on RT–qPCR 72 h after p38 depletion (Fig 3C), suggesting that p38 plays a key role in *MYC* mRNA regulation but that another pathway downstream from FGFR3 is probably responsible for regulating the stability of the protein (Fig 1).

MYC degradation by the proteasome is regulated by glycogen synthase kinase 3 (GSK3). The activity of GSK3 is regulated by phosphorylation, including that of the Ser9 residue of GSK3b and the Ser21 residue of GSK3a, by AKT, in particular (Gregory *et al*, 2003). In accordance with this mechanism, we demonstrated, in RT112 and MGH-U3 cells, that FGFR3 inhibition with a pan-FGFR inhibitor (PD173074) decreased the phosphorylation of AKT at Ser473 and that of GSK3β at Ser9 (Fig 3D). We found that PI3-kinase inhibition by LY294002 inhibited AKT phosphorylation and decreased both the phosphorylation of the Ser9 residue of GSK3β and MYC protein levels (Fig 3E). Thus, FGFR3 induces AKT phosphorylation, leading to the inhibition of GSK3β through Ser9 phosphorylation, thereby preventing the proteasome-mediated proteolysis of MYC. Our results thus demonstrate that, downstream from the aberrantly activated FGFR3, both p38 and AKT are involved in the induction of MYC accumulation, which, in turn, drives cell proliferation. Consistent with these results for p38, reverse-phase protein array (RPPA) analysis of a panel of 129 tumors showed that p38 was significantly more phosphorylated in tumors expressing a mutated FGFR3 than in tumors expressing wild-type FGFR3 (Fig 3F, left panel). AKT was not differentially phosphorylated in tumors with and without *FGFR3* mutations, suggesting that, in bladder cancer, AKT can be activated by several mechanisms including the aberrant activation of FGFR3,

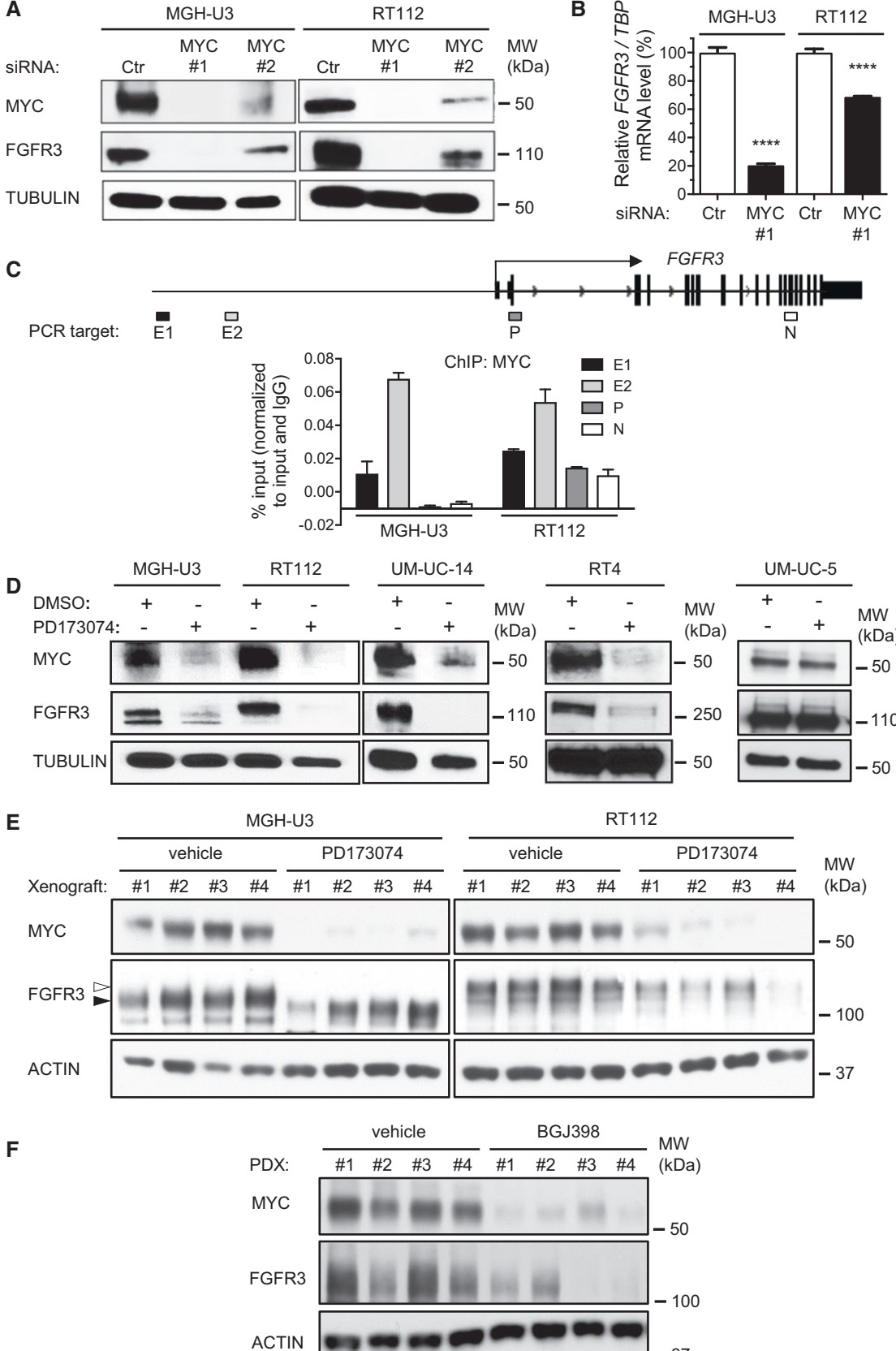

**Figure 2.**

◄

**Figure 2.  MYC and FGFR3 are involved in a positive feedback loop in bladder cancer cell lines expressing an activated form of FGFR3.**

A  The expression of MYC and FGFR3 was analyzed by Western blotting in lysates from MGH-U3 and RT112 cells transfected for 72 h with *MYC* siRNAs. Tubulin was used as a loading control.

B  Relative *FGFR3* mRNA levels in MGH-U3 and RT112 cells transfected for 72 h with siRNAs targeting *MYC* or a control siRNA (Ctr). The results presented are the means of two independent experiments carried out in triplicate; the standard errors are indicated. Unpaired Student's *t*-tests were used for comparison with the control, ****$P < 0.0001$.

C  ChIP–qPCR for MYC at the *FGFR3* locus in MGH-U3 and RT112 cells (lower panel). The qPCR target loci of *FGFR3* are schematized (upper panel). Data presented are representative of two replicate experiments. Error bars show standard deviation of three replicate qPCR reactions.

D  RT112, MGH-U3, UM-UC-14, RT4, and UM-UC-5 cells were treated for 48 h with a pan-FGFR inhibitor (500 nM PD173074). Lysates were obtained, and levels of FGFR3 and MYC were analyzed by Western blotting with appropriate antibodies. An anti-tubulin antibody was used as a loading control.

E  MGH-U3 and RT112-derived xenograft tumors from mice treated for 9 days with vehicle or PD173074 (25 mg/kg/day) were lysed and immunoblotted with anti-FGFR3 and anti-MYC antibodies. Actin was used as a loading control. Black and white arrowheads indicate WT FGFR3 and FGFR3-TACC3 bands, respectively.

F  PDX tumors bearing the FGFR3-S249C mutation from mice treated for 4 days with vehicle or BGJ398 (30 mg/kg/day) were lysed and immunoblotted with anti-FGFR3 and anti-MYC antibodies. Actin was used as a loading control.

Source data are available online for this figure.

such as EGFR activation in basal tumors (Rebouissou *et al*, 2014; Fig 3F, right panel). The FGFR3/MYC positive feedback loop involving p38 and AKT activation by FGFR3 identified in bladder cancer cell lines may, therefore, also occur in human bladder tumors with genetic alterations of FGFR3. The disruption of this loop with inhibitors of AKT and p38 may, therefore, constitute an effective way of treating these tumors.

**Targeting FGFR3, p38, or AKT is an effective strategy for inhibiting the growth and transformation of bladder cancer cells expressing aberrantly activated FGFR3**

We evaluated the effects of p38 and PI3K inhibitors (SB203580 and LY294002, respectively) on the viability of RT112 and MGH-U3 cells (Fig 4A) and on MGH-U3 cell transformation (Fig 4B). The inhibition of these two pathways decreased the viability of MGH-U3 and RT112 cells and the anchorage-independent growth of MGH-U3 cells as efficiently as FGFR3 inhibition with a pan-FGFR inhibitor, PD173074. Using a *MAPK14* siRNA, we confirmed that p38α depletion decreased the viability of RT112 and MGH-U3 cells and the anchorage-independent growth of MGH-U3 cells (Fig 4C and D). We also validated *in vivo* the critical role of p38 in mutated FGFR3-induced tumor growth, by showing that p38 inhibition with SB203580 significantly slowed the tumor growth of MGH-U3 and RT112 xenografts in athymic nude mice (Fig 4E). An AKT inhibitor has already been shown to decrease MGH-U3 xenograft growth slightly in athymic nude mice (Davies *et al*, 2015).

**MYC acts as a key master regulator of proliferation in the FGFR3 pathway, rendering FGFR3-dependent cells sensitive to a BET bromodomain inhibitor (JQ1)**

We looked for other ways to disrupt the FGFR3/MYC loop in bladder tumors bearing *FGFR3* mutations. Recent studies have shown that the indirect inhibition of MYC through the targeting of proteins involved in the regulation of its transcription is an effective strategy for treating MYC-dependent tumors (Posternak & Cole, 2016). In particular, several studies have highlighted the use of bromodomain inhibitors as an effective strategy for blocking *MYC* transcription (Delmore *et al*, 2011; Mertz *et al*, 2011). We therefore focused on JQ1, a potent and well-characterized BET bromodomain inhibitor that inhibits the binding of bromodomain-containing protein 4 (BRD4) to acetylated lysine residues on histones, thereby preventing

transcription. It is particularly active against *MYC*, the transcription of which seems to be dependent on the binding of BRD4 to its enhancers or "super-enhancers" (Lovén *et al*, 2013). We first analyzed the BRD4 occupancy of the *MYC* locus by ChIP–qPCR in the RT112 and MGH-U3 bladder cell lines (Fig 5A). Using publicly available data for histone marks, we designed primers binding to one potential enhancer, one control negative region and the promoter (Appendix Fig S8A) and checked that the selected regions harbored the expected histone marks in the RT112 bladder cell line (Appendix Fig S8B). The *MYC* enhancer was slightly enriched in BRD4, and the *MYC* promoter was strongly enriched in BRD4. In both cases, this enrichment was prevented by JQ1 treatment (Fig 5A). We then checked, by Western blotting, that JQ1 treatment inhibited MYC and FGFR3 expression, in both cell lines (Fig 5B). The observed inhibition was of similar strength to the FGFR3 inhibition observed with 1 μM PD173074 (Fig 5B). By contrast, treatment with (−)-JQ1, the inactive enantiomer of (+)-JQ1, had little impact on MYC and FGFR3 levels (Fig 5B). Consistent with this inhibition of MYC and FGFR3 expression following JQ1 treatment, we also showed that JQ1 treatment significantly decreased the viability of RT112 and MGH-U3 cells *in vitro* (Fig 5C). Finally, we showed *in vivo* that JQ1 treatment significantly slowed the growth of MGH-U3 and RT112 xenografts in nude mice (Fig 5D). However, on the one hand, the inhibition of tumor growth by JQ1 treatment was relatively modest. In the other hand, although they slowed tumor growth, FGFR inhibitors did not trigger a regression of tumor size (Appendix Fig S6A). We therefore hypothesized that a combinatorial treatment might improve the response. We tested this hypothesis *in vitro,* on MGH-U3 and RT112 cell viability that made it possible to use ranges of doses for both molecules (Appendix Fig S9A). We found that simultaneous use of the two drugs increased treatment efficacy over that achieved with the two drugs used separately, as highlighted in Fig 5E. A mathematical analysis of our results by the Loewe additivity method (Foucquier & Guedj, 2015) showed that the two drugs had an additive effect in most cases, and possibly even a synergistic effect at some concentrations (Appendix Fig S9B).

# Discussion

Alterations of FGFR3 (mutations or translocation) are among the most frequent genetic events in bladder carcinoma, occurring in

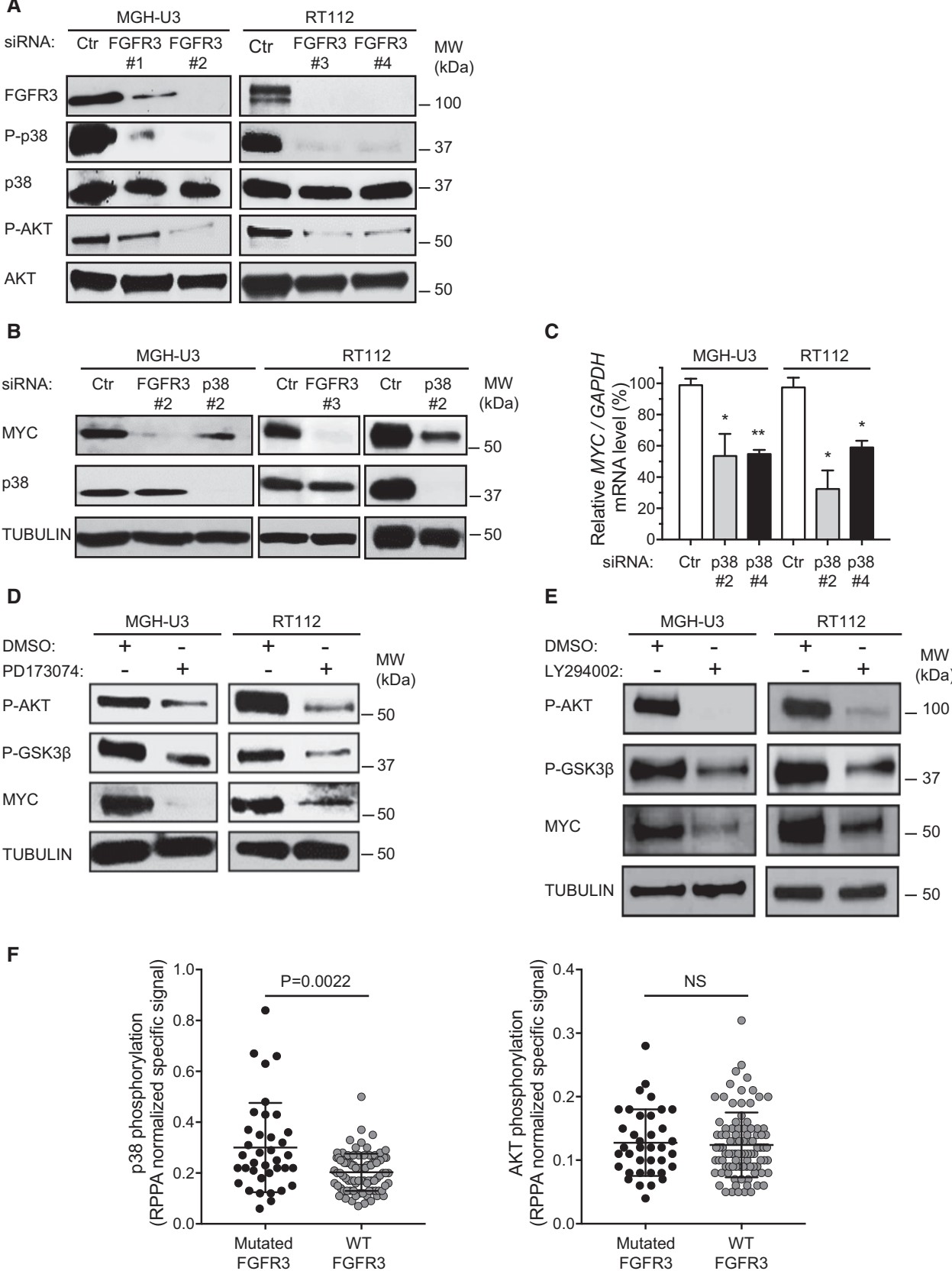

Figure 3.

**Figure 3.  The MYC accumulation induced by activated FGFR3 is dependent on the activation of p38 and AKT.**

A   MGH-U3 and RT112 human bladder tumor cells were transfected with control siRNA (Ctr) or with siRNAs targeting *FGFR3*. Lysates were obtained and the levels of p38, phospho-p38 [P-p38 Thr180/Tyr182], AKT, phospho-AKT (P-AKT Ser473)], and FGFR3 were assessed by Western blotting. Different siRNAs were used in the two cell lines (see Materials and Methods).

B   MGH-U3 and RT112 cells were transfected for 72 h with a siRNA targeting either *FGFR3* or *MAPK14* (p38α). Lysates were obtained and MYC and p38 protein levels were analyzed by Western blotting.

C   MGH-U3 and RT112 cells were transfected with *MAPK14* (p38α) siRNA for 72 h. The level of *MYC* mRNA level was determined by RT–qPCR (left panel). The results presented are the means and standard errors of two independent experiments carried out in triplicate. Unpaired Student's *t*-tests were used for comparison with appropriate siRNA control (Ctr), *P < 0.05; **P < 0.005.

D   MGH-U3 and RT112 cells were treated for 2 h with DMSO or a FGFR inhibitor (0.5 µM PD173074). Lysates were obtained and analyzed by Western blotting with antibodies against MYC, phospho-AKT (Ser473) and phospho-GSK3β (Ser9). Tubulin was used as a loading control.

E   Western blot comparing MYC, phospho-AKT (Ser473) and phospho-GSK3β (Ser9) levels in MGH-U3 and RT112 cells treated for 3 h with a PI3 kinase inhibitor (20 µM LY294002) or control DMSO. Tubulin was used as a loading control.

F   The level of phosphorylation of p38 (left panel) and AKT (right panel) was assessed by reverse-phase protein array (RPPA) in 129 human bladder tumors, as described in the Materials and Methods. *FGFR3* mutations were present in 38 tumors. No tumor harbored an FGFR3-TACC3 or FGFR3-BAIAP2L1 fusion gene. Mann–Whitney test was used for comparisons between mutated and non-mutated tumors. Means and standard errors are represented.

about 70% of NMIBCs and 20% of MIBCs. These alterations induce the constitutive activation of FGFR3 and lead to an oncogene dependence to FGFR3. In this study, we characterized further the mechanisms involved in the activity of aberrantly activated FGFR3, highlighting new possibilities for the treatment of bladder tumors with activating alterations of FGFR3. We found that MYC played a crucial role in the aberrantly activated FGFR3 pathway. This transcription factor regulated by FGFR3 was involved in FGFR3-driven cell proliferation in two bladder cancer-derived cell lines expressing FGFR3 (FGFR3-Y375C) or the fusion protein FGFR3-TACC3. We also showed that MYC upregulated FGFR3 expression directly, by binding to enhancers upstream from FGFR3, as part of a FGFR3/MYC positive feedback loop operating both *in vitro* and *in vivo* in bladder cancer-derived cell lines xenografts and in a PDX model bearing an FGFR3 mutation. The FGFR3-driven accumulation of MYC was due to both an increase in *MYC* mRNA levels and stabilization of the MYC protein. FGFR3 increases *MYC* mRNA levels by activating the p38α MAP kinase. FGFR3 also induces stabilization of the MYC protein, by activating AKT, which, in turn, phosphorylates the Ser9 residue of GSK3β, thereby preventing its interaction with MYC and the degradation of this protein by the proteasome. Finally, our results provide *in vitro* and *in vivo* proof of concept in xenografts that the inhibition of MYC expression, and, in turn, of FGFR3 expression, by an inhibitor of AKT or p38 or a BET bromodomain inhibitor (JQ1) is a potentially effective strategy for the treatment of FGFR3-dependent bladder tumors. The results obtained with FGFR3 inhibitors in one PDX model and in two cell lines xenografts were similar, increasing our confidence in the relevance of our results to human tumors. Based on the results presented here, we devised a model for this newly identified FGFR3/MYC positive feedback loop involved in bladder tumor cell proliferation (Graphical abstract). Interestingly, studies of *MYC* mRNA levels and of the phosphorylation of p38 and AKT in human bladder tumor samples harboring *FGFR3* mutations suggested that this loop might also operate in tumors. The relevance to human tumors was further supported by the decrease in FGFR3 and MYC levels following anti-FGFR treatment in a PDX model bearing an *FGFR3* mutation. The insight into the aberrantly activated FGFR3 pathway provided by this study could make it possible to identify tumors presenting alterations to this pathway, such as MYC overexpression or p38 activation, in the absence of FGFR-activating genetic alterations. These tumors might also benefit from the alternative therapeutic strategies proposed in this study for bladder tumors displaying aberrant FGFR3 activation.

We found that FGFR3 activation increased MYC expression and that *FGFR3* was a direct transcriptional target of MYC. This FGFR3/MYC positive feedback loop probably contributes to the higher levels of FGFR3 expression previously observed in human bladder tumors with *FGFR3* mutations (Bernard-Pierrot *et al*, 2006). It has been suggested that this overexpression is also mediated by a loss of microRNAs99/100 targeting *FGFR3* in bladder tumors (Catto *et al*, 2009; Blick *et al*, 2013).

In this study, we searched for transcriptional regulators involved in the regulation of gene expression induced by two types of aberrantly activated FGFR3: a mutated form of the receptor (Y375C) and a fusion protein (FGFR3-TACC3). Specific signaling pathways—PLCγ activation (Williams *et al*, 2013) and localization to the kinetochore (Singh *et al*, 2012)—have been associated with these two forms, but we observed a large overlap between the transcriptional regulators driven by these two types of receptors. Furthermore, both types of receptor acted via the same molecular mechanism, the activation of p38 and AKT, leading to MYC accumulation, resulting in the induction of hyperproliferation. Most of the upstream regulators activated by both types of receptor in this study have also been shown to be regulated by FGFR3-BAIAP2L1, another form of aberrantly activated FGFR3, in RAT2 cells (Nakanishi *et al*, 2015). In both this and a previous study, we found that FGFR3 activation inhibited tumor suppressor pathways involving RB1/RBL1, TP53, or P16 (*CDKN2A*) and activated pro-proliferative pathways involving E2F, CCND1, or TBX2. However, the MYC activation described here was not observed in RAT2 cells. This discrepancy may reflect differences in the technical approach used (inhibition versus overexpression), the species and tissues studied (human epithelium versus rat fibroblast) or the thresholds used to identify genes regulated by FGFR3, and the upstream regulators involved in their regulation.

In a study using a very different approach published during the preparation of this manuscript, MYC was also implicated in pathways involving activated FGFRs in several different types of cancer (Liu *et al*, 2016). This study showed that the altered FGFRs were associated with an increase in MYC protein stability. In one cell line displaying FGFR1 amplification, the authors showed, as suggested by our data for FGFR3 revealing a lack of synergy between *MYC* and *FGFR3* knockdown, that MYC was the main effector of FGFR1

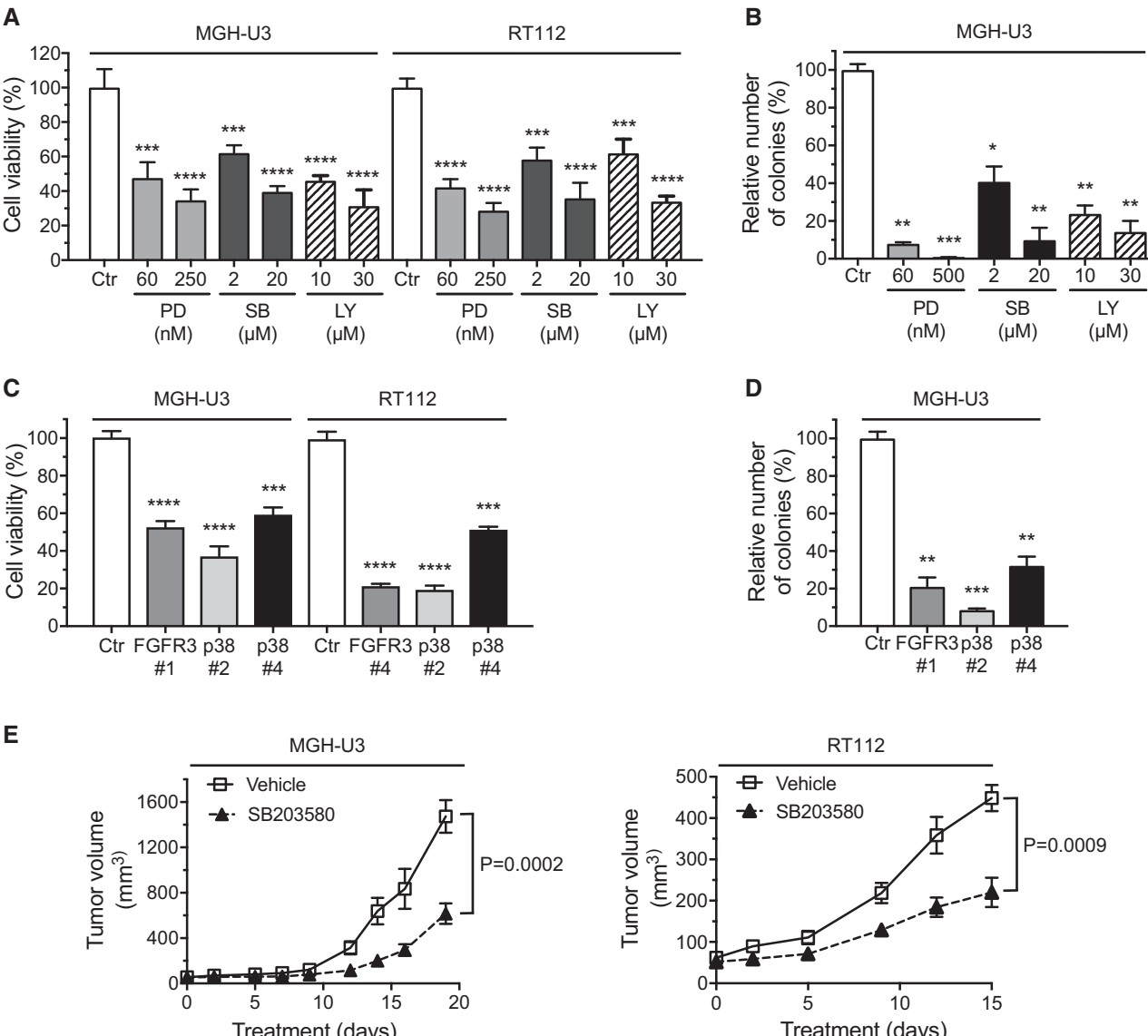

**Figure 4.  The inhibition of p38 or AKT reduces the growth and transformation of bladder cancer cells expressing aberrantly activated FGFR3.**

A  MGH-U3 and RT112 cells were treated with control DMSO, PD [PD173074 (FGFR inhibitor)], SB [SB203580 (p38 inhibitor)] or LY [LY294002 (PI3 kinase inhibitor)] for 72 h and cell viability was then assessed by measuring MTT incorporation.

B  Impact of PD (PD173074), SB (SB203580), or LY (LY294002) treatment on the cell anchorage-independent growth of MGH-U3 cells. Colonies in soft agar with diameters greater than 50 μm were counted 14 days after seeding in the presence of inhibitors.

C  Comparison of the effects of *MAPK14* (p38α isoform) and *FGFR3* knockdown on the viability of MGH-U3 and RT112 cells, as measured by MTT incorporation.

D  Soft agar colony formation assay for MGH-U3 cells transfected with siRNA against *FGFR3* or *MAPK14* (p38α isoform). Cells were grown for 14 days before counting.

E  MGH-U3 bladder cancer cells were injected into nude mice (*n* = 5 animals/group), two xenografts per animal (one in each flank). Nine days later, the mice received an injection of vehicle or SB203580 (100 μl of 20 μM SB203580) into the tumor, once daily, 6 days per week. Tumor size was measured at the indicated time point, and tumor volume was calculated.

Data information: (A–D) The results presented are the means of two independent experiments carried out in triplicate; the standard errors are indicated. Unpaired Student's *t*-tests were used to assess the significance of differences, *P < 0.05; **0.001 < P < 0.005; ***0.0001 < P < 0.001; ****P < 0.0001. (E) Data are presented as means ± SEM. Results were compared in Mann–Whitney test.

activity, because the effect of FGFR inhibitors was abolished by an undegradable MYC mutant. They suggested that MYC protein stabilization was due to the activation of ERK1/2, rather than AKT as described here. Our results clearly highlighted the crucial role of AKT in sustaining MYC stability through the phosphorylation of

GSK3β. However, we did not study the impact of ERK1/2 inactivation in our FGFR3-dependent bladder tumor models and we cannot, therefore, rule out the possible involvement of this pathway in cooperation with the AKT pathway, as shown for RAS (Sears *et al*, 2000; Yeh *et al*, 2004). Furthermore, we also highlighted the role of

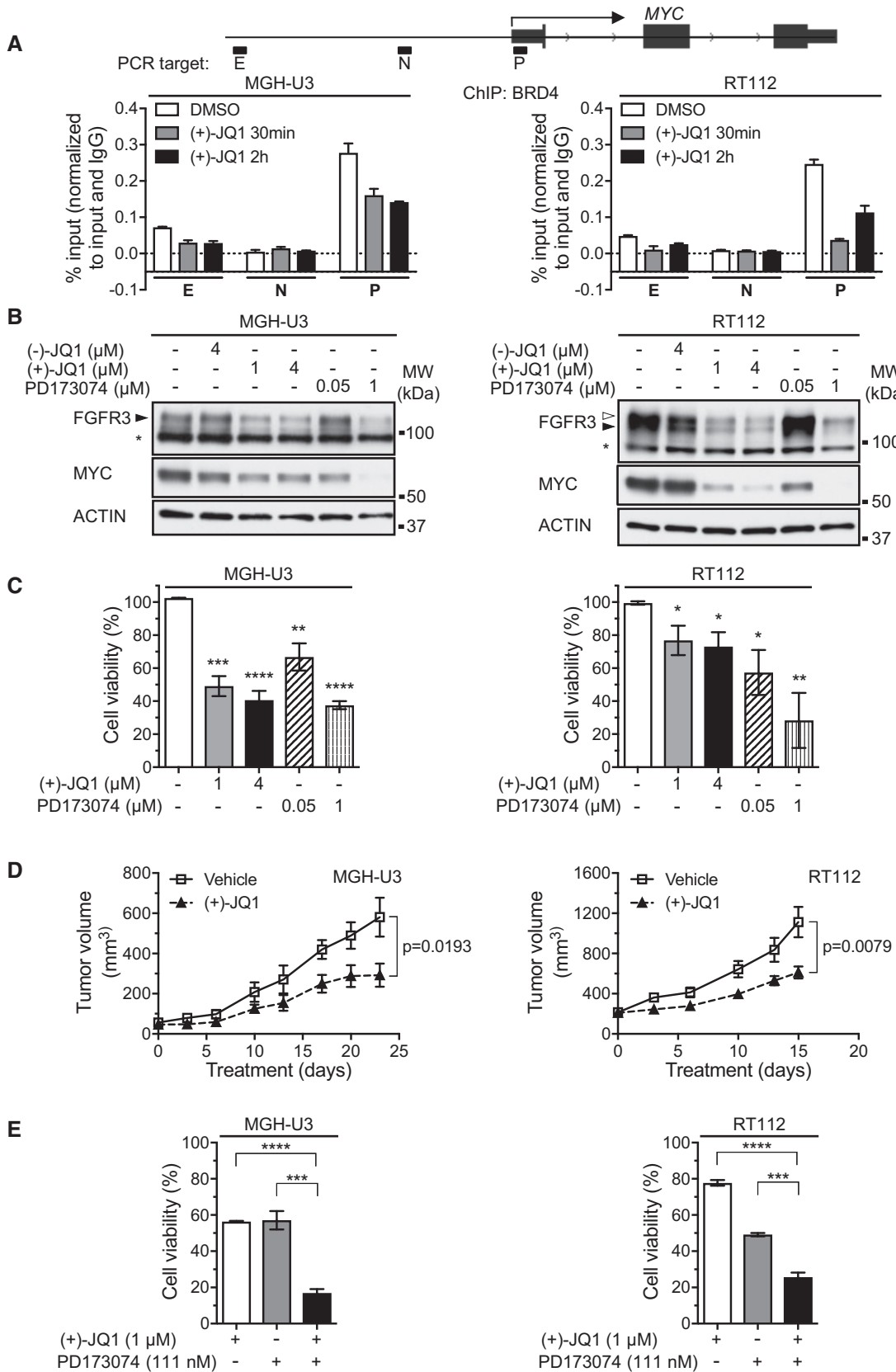

**Figure 5.**

**Figure 5.  The MYC accumulation induced by activated FGFR3 confers sensitivity to BET bromodomain inhibitors in FGFR3-dependent bladder cancer cells *in vitro* and *in vivo*.**

A   The qPCR target loci for *MYC* are shown (upper panel). ChIP–qPCR of BRD4 for the *MYC* locus in MGH-U3 and RT112 cells treated with DMSO or 1 μM (+)-JQ1 for 30 or 120 min (lower panels). Data presented are representative of two replicate experiments. Error bars show standard deviation of three replicate qPCR reactions.

B   Western blot analysis of MYC and FGFR3 expression in lysates from MGH-U3 and RT112 cells treated with (+)-JQ1 (1 or 4 μM) for 48 h. Anti-actin antibody was used as a loading control. Pan-FGFR inhibitor, PD173074 (50 nM and 1 μM), and inactive enantiomer (−)-JQ1 (4 μM) were used as controls. Black and white arrowheads indicate WT FGFR3 and FGFR3-TACC3 bands, respectively. Asterisk indicates non-specific band.

C   MGH-U3 and RT112 cells were treated for 72 h with DMSO, (+)-JQ1 (1 or 4 μM) or PD133074 (50 nM or 1 μM). Cell viability was measured with CellTiter-Glo. Results were compared with those for a DMSO control in unpaired Student's *t*-tests, *$P < 0.05$; **$0.001 < P < 0.005$; ***$0.0001 < P < 0.001$; ****$P < 0.0001$. Means and standard errors are represented. Three replicates were performed.

D   MGH-U3 and RT112 bladder cancer cells were injected into nude mice ($n = 6$ animals/group), two xenografts per animal (one in each flank). Nine and seven days later, the mice received an injection of vehicle or (+)-JQ1 (IP injection, 50 mg/kg, once daily, 6 days per week), respectively. Tumor growth was assessed twice weekly, by measuring tumor size. Data are presented as means ± SEM. Results were compared in Wilcoxon's test.

E   MGH-U3 and RT112 cells were treated for 72 h with (+)-JQ1 and PD133074 alone or in combination. Cell viability was measured with CellTiter-Glo. Data are presented as means ± SD of three experiments carried out in triplicate. Results for the drug combination were compared with those for each individual drug separately, in unpaired Student's *t*-tests, ***$0.0001 < P < 0.001$; ****$P < 0.0001$.

Source data are available online for this figure.

mutated FGFR3, dependent on p38 activation, in the upregulation of *MYC* mRNA levels both in cell lines and in a PDX model. MYC overexpression which often leads to MYC oncogene addiction has been associated with aggressive phenotype in many tumor types (Dang, 2012; Stine *et al*, 2015). This is not the case in bladder cancer since the majority of FGFR3-mutated tumors are low-stage, low-grade tumors. Furthermore, among FGFR3-mutated tumors, no difference in MYC expression could be observed in MIBC and NMIBC (data not shown). This could be related to a specific FGFR3-induced MYC transcriptomic program in bladder tumors (Kress *et al*, 2015). Consistent with its key role in FGFR signaling, MYC was also recently identified as a potential marker of the anti-FGFR response, because cells expressing both MYC and FGFRs have been shown to be more sensitive to anti-FGFR therapies (Malchers *et al*, 2014; Liu *et al*, 2016). In light of its key role downstream from FGFR, MYC inhibition appears to be a valuable therapeutic strategy for bladder tumors with FGFR3 alterations. MYC has emerged as a clear therapeutic target in other cancers, and many strategies for inhibiting MYC activity through direct or indirect means have been described (Posternak & Cole, 2016). We evaluated the therapeutic potential of BET bromodomain inhibitors, a class of epigenetic modulators that emerged in a clinical setting. We demonstrated that JQ1 prevented BRD4 binding to the *MYC* promoter and enhancer, thereby inhibiting *MYC* expression and, consequently, the growth of bladder tumor cells expressing activated forms of FGFR3 both *in vitro* and *in vivo* in xenograft. These preclinical results suggest that bladder tumors with FGFR3 alterations could potentially be treated with BET bromodomain inhibitors. Resistance to monotherapy with BET bromodomain inhibitors has been observed and linked to kinome reprogramming in ovarian cancer (Kurimchak Alison *et al*, 2016) or to a decrease in PP2A activity in triple-negative breast cancer (Shu *et al*, 2016). Resistance to anti-FGFR therapies has also been observed in FGFR3-dependent cells and linked to the activation of ERBB2/3 or EGFR (Herrera-Abreu *et al*, 2013; Wang *et al*, 2015). These observed resistances could be overcome by combination strategies, involving PI3K inhibitors, for example (Wang *et al*, 2017). The use of a combination of a pan-FGFR inhibitor and a BET bromodomain inhibitor induced a stronger growth inhibition as compared to each individual drug *in vitro*. *In vivo* tests for these treatments are currently underway for our PDX model.

We also demonstrated that the activation of both p38 and AKT was critical for the induction of bladder cancer cell proliferation and transformation by FGFR3. This critical role was linked to the ability of these two pathways to induce MYC accumulation, by increasing *MYC* mRNA levels and by stabilizing the MYC protein, respectively. The role of AKT in cancer progression has been clearly demonstrated for various tumors (Vivanco & Sawyers, 2002), including bladder cancer (Calderaro *et al*, 2014). The role of p38 in cancer is dual, p38 playing both a tumor suppressor role by inducing cell apoptosis and protumorigenic functions depending on the cancer types (Koul *et al*, 2013; Igea & Nebreda, 2015). The opposite functions could be related to cell specificity, nature of the stimuli, the isoform activated since p38 exist as four isoforms or the component downstream p38. p38 can be activated by tyrosine kinase receptors (PDGF receptor, VEGF receptor, EGF receptor, FGFR1) to function as a positive regulator of tumor progression, mediating motility and invasion, suppressing apoptosis, stimulating the epithelial-to-mesenchymal transition in various cell types (Bates & Mercurio, 2003; Frey *et al*, 2004; Nishihara *et al*, 2004). Our results demonstrating that p38α promotes proliferation, by upregulating *MYC* mRNA levels are in line with the protumorigenic functions of p38 and with recent studies in breast, head and neck cancers and nasopharyngeal carcinoma (Leelahavanichkul *et al*, 2014; Li *et al*, 2017; Wada *et al*, 2017). The activation of p38 by activated-FGFR3 in bladder tumors contributes to malignant behavior and the inhibition of this activation may be of therapeutic value, as reported for an increasing number of cancers (Koul *et al*, 2013; Igea & Nebreda, 2015).

Our results thus suggest alternative strategies targeting different aspects of FGFR3 signaling that might be beneficial for the treatment of bladder tumors expressing aberrantly activated FGFR3. Targeting two parts of the signaling pathway simultaneously may increase treatment efficacy or delay the development of tumor resistance, as observed clinically in melanomas harboring *RAF* mutations managed with treatments targeting RAF and MEK (Flaherty *et al*, 2012). It has already been reported that of the simultaneous inhibition of FGFR3 and AKT in MGH-U3 xenograft models increases treatment efficacy over than achieved with either of the two drugs used separately (Davies *et al*, 2015). Such strategies are widely tested in different tumor types and in

particular using pan-FGFR inhibitors. A multi-drug phase II clinical trial including pan-FGFR inhibitor (BGJ398) together with a MEK inhibitor (MEK162) and a RAF inhibitor (LGX818) is currently ongoing in advanced BRAF melanoma (NCT02159066). A phase Ib trial of BGJ398 in combination with BYL719 (PI3K inhibitor) on solid tumors showed encouraging results as eight patients over 24 showed a partial response, among them, one patient with a urothelial carcinoma bearing FGFR3-TACC3 had a complete tumor shrinkage for 4 months (NCT01928459). FGFR3-TACC3 fusion protein expression has been reported in several other cancers, including glioblastoma (Singh *et al*, 2012) and lung adenocarcinoma (Capelletti *et al*, 2014). It would be interesting to determine whether this FGFR3/MYC feedback loop, mediated by AKT and p38, also operates in other types of human cancers expressing FGFR3-TACC3. If so, these treatments could be extended to other cancer types.

## Materials and Methods

### Cell culture and transfection

The human bladder-derived cell lines RT112, RT4, UM-UC-14, and UM-UC-5 were obtained from DSMZ (Heidelberg, Germany). MGH-U3 cells were kindly provided by Dr. Paco Real (CNIO, Madrid). We mostly used RT112 and MGH-U3 cells. RT112 cells were derived from a transitional cell carcinoma (TCC; histological grade G2) excised from a woman with untreated primary urinary bladder carcinoma. The MGH-U3 cell line was established with cells from a 76-year-old patient with a history of recurrent non-invasive bladder carcinomas (papillary TCC, histological grade G1; Lin *et al*, 1985). MGH-U3 cells harbor a homozygous FGFR3-Y375C mutation and RT112 cells have a FGFR3-TACC3 translocation. A comprehensive genomic characterization of these cells has been reported (Earl *et al*, 2015). MGH-U3, UM-UC-5, and UM-UC-14 cells were cultured in DMEM, whereas RT112 and RT4 cells were cultured in RPMI. Media were supplemented with 10% fetal calf serum (FCS). Cells were incubated at 37°C, under an atmosphere containing 5% $CO_2$. The identity of the cell lines used was checked by analyzing genomic alterations with comparative genomic hybridization arrays (CGH array), and the *FGFR3* and *TP53* mutations were checked with the SNaPshot technique (for *FGFR3*) or by classical sequencing (for *TP53*), the results obtained being compared with the initial description of the cells. We routinely checked for mycoplasma contamination.

Transfected NIH-3T3 cells expressing the mutated human FGFR3b-S249C receptor (clones S249C1.1, S249C 1.2) or transfected with the control pcDNAI-Neo plasmid (clones Neo1.5, Neo 2.1) were established during a previous study (Bernard-Pierrot *et al*, 2006). They were cultured in DMEM supplemented with 10% newborn calf serum (NCS), 2 mM glutamine, 100 U/ml penicillin, 100 μg/ml streptomycin, and 400 μg/ml G418.

For siRNA transfection, MGH-U3 and RT112 cells were used to seed six-well or 24-well plates at a density of 250,000 cells/well for MGH-U3 cells and 200,000 cells/well for RT112 cells. Cells were transfected with 5 (*FGFR3* siRNA #3 and #4) or 20 nM siRNA in the presence of Lipofectamine RNAi Max reagent (Invitrogen), in accordance with the manufacturer's protocol. siRNAs were purchased

from Ambion and Qiagen. For the control siRNA, we used a Qiagen control targeting luciferase (SI03650353).

The sequences of the siRNAs were as follows:

| *FGFR3* #1 | 5′-GCUUUACCUUUUAUGCAA-3′ (sense strand) |
|---|---|
| | 5′-UUGCAUAAAAGGUAAAGGC-3′ (antisense strand) |
| *FGFR3* #2 | 5′-GGGAAGCCGUGAAUUCAGU-3′ (sense strand) |
| | 5′-ACUGAAUUCACGGUUCCC-3′ (antisense strand) |
| *FGFR3* #3 | 5′-CCGUAGCCGUGAAGAUGC-3′ (sense strand) |
| | 5′-AGCAUCUUCACGGCUACGG-3′ (antisense strand) |
| *FGFR3* #4 | 5′-CCUGCGUCGUGGAGAACA-3′ (sense strand) |
| | 5′-UUGUUCUCCACGACGCAGG-3′ (antisense strand) |

*FGFR3* siRNA#1 and siRNA#2 targeted exon 19 of *FGFR3* (NM_001163213). They therefore knocked down the expression of wild-type and mutated FGFR3, but not of the FGFR3-fusion gene containing the first 18 exons of FGFR3 (Wu *et al*, 2013). Conversely, siRNA#3 and siRNA#4 targeted exons 12 and 6 of *FGFR3* (NM_001163213), respectively, knocking down both wild-type and FGFR3-TACC3 expression in RT112 cells.

| p38α #2 | 5′-GGUCUCUGGAGGAAUUCAA-3′ (sense strand) |
|---|---|
| | 5′-UUGAAUUCCUCCGAGACC-3′ (antisense strand) |
| p38α #4 | 5′-CUGCGGUUACUUAAACAUA-3′ (sense strand) |
| | 5′-UAUGUUUAAGUAACCGCAG-3′ (antisense strand) |
| p38α refers to *MAPK14* | |
| MYC #1 | 5′-UCCCGGAGUUGGAAAACAATT-3′ (sense strand) |
| | 5′-UUGUUUUCCAACUCCGGGATC-3′ (antisense strand) |
| MYC #2 | 5′-CGGUGCAGCCGUAUUUCUATT-3′ (sense strand) |
| | 5′-UAGAAAUACGGCUGCACCGAG-3′ (antisense strand) |

Cell viability was assessed with the MTT assay (0.5 mg/ml) in 24-well plates, or the CellTiter-Glo assay (Promega) in 96-well plates, 72 h after transfection. Cell lysates were also prepared 72 h after transfection, in six-well plates, for subsequent immunoblotting analysis.

### Kinase and protein inhibitors

The inhibitors LY294002, PD98059, SB203580, SU5402, and PD173074 were purchased from Calbiochem (Merck Eurolab, Fontenay Sous Bois, France). MG132 was obtained from Selleckchem (Euromedex, Souffelweyersheim, France). BGJ398 was purchased from LC Laboratories (USA).

The inhibitors (+)-JQ1, (−)-JQ1 and PD173074 (for *in vivo* studies) were purchased from MedChem Express (MedChemtronica, Stockholm, Sweden).

### Immunoblotting

NIH-3T3, MGH-U3, RT112, RT4, UM-UC-14, and UM-UC-5 cells were resuspended in Laemmli lysis buffer [50 mM Tris-HCl (pH 6.8), 2 mM DTT, 2.5 mM EDTA, 2.5 mM EGTA, 2% SDS, 5% glycerol with protease inhibitors and phosphatase inhibitors

(Roche)], and the resulting lysates were clarified by centrifugation. The protein concentration of the supernatants was determined with the BCA protein assay (Thermo Scientific, France). Proteins (10–50 μg) were resolved by SDS–PAGE in 10% polyacrylamide gels, electrotransferred onto Bio-Rad nitrocellulose membranes, and analyzed with antibodies against p38 and the phosphorylated form of p38 (Thr180/Tyr182; Cell Signaling Technology # 9212 and # 4511, used at 1/5,000), AKT and the phosphorylated form of AKT (Ser473; Cell Signaling Technologies # 2920 and # 4060, used at 1/5,000), GSK3β (Ser9; Cell Signaling Technology # 5558, used at 1/1,000), MYC (Cell Signaling Technology # 9402, used at 1/1,000), α-tubulin and β-actin (Sigma Aldrich #T6199, used at 1/15,000 and #A2228, used at 1/25,000), or the extracellular domain of FGFR3 (Abcam, # ab133644, 1/5,000). Anti-mouse IgG, HRP-linked, and anti-rabbit IgG, HRP-linked antibody (Cell Signaling Technology # 7076 and # 7074, used at 1/3,000) were used as secondary antibodies. Protein loading was checked by Amido Black staining of the membrane after electrotransfer.

### ChIP–qPCR

RT112 and MGH-U3 cells were cross-linked directly by adding 1% formaldehyde to the medium and incubating for 10 min at room temperature. The reaction was stopped by adding glycine to a final concentration of 0.125 M and incubating for 5 min at room temperature. The cells were then harvested. The fixed cells were rinsed twice with PBS, resuspended in extraction buffer [0.25 M sucrose, 10 mM Tris–HCl pH 8, 10 mM $MgCl_2$, 1% Triton, 5 mM β−mercaptoethanol, protease inhibitors (Roche)] and centrifuged at 3,000 × $g$ for 10 min. We then used the ChIP-IT® High Sensitivity kit (Active motif), treating the samples according to the manufacturer's instructions. ChIP was performed with the following antibodies: mouse anti-BRD4 (Bethyl Laboratories A301-985A50), rabbit polyclonal anti-c-MYC (Santa Cruz sc-764), anti-H3K4me3 (Abcam ab8580-25), and anti-H3K27ac (Abcam ab4729) antibodies and the rabbit IgG polyclonal isotype control antibody (Abcam ab37415).

For ChIP–qPCR experiments, quantitative PCR was performed with the SYBR Green PCR kit from Applied Biosystems. Enrichment in ChIPed DNA was calculated as a percentage of the input minus IgG ChIP signal. The sequences of the primers used were as follows:

| FGFR3 locus | | |
|---|---|---|
| E1 | AAGATGAGCAAGGCACCTG (forward) | CTCCAGGTCAGAACCAAAGC (reverse) |
| E2 | ACACGCAGGCACACACAG (forward) | AGGGCTTGTTGCTTCCTCTG (reverse) |
| P | GCAGGTAAGAAGGGACCCAC (forward) | CGGAATCCGGGCTCTAACC (reverse) |
| N | ACTCCTTCGACACCTGCAAG (forward) | GTCCTTGAAGGTGAGCTGCT (reverse) |
| MYC locus | | |
| E | TCTTGCCAGACCTAATGCTG (forward) | CCTTGGCCACATTGCTTATC (reverse) |
| N | CAGCTAAATGGCACATAGGC (forward) | ATATTGCCCCGGCTAATCTC (reverse) |
| P | TTCGGGTAGTGGAAAACCAG (forward) | GTGTCAATAGCGCAGGAATG (reverse) |

### Soft agar assay

MGH-U3 cells (20,000), untransfected or transfected with siRNA, were used to seed 12-well plates containing DMEM supplemented with 10% FCS and 1% agar, in triplicate. Cells were cultured in the presence or absence of inhibitors in the agar and culture medium, as appropriate. The medium was changed weekly. The plates were incubated for 14 days, and colonies larger than 50 μm in diameter, as measured with a phase-contrast microscope equipped with a measuring grid, were counted.

### RNA extraction from cell lines

RNA was isolated from cell lines with RNeasy Mini kits (Qiagen, Courtaboeuf, France).

### Real-time reverse transcription-quantitative PCR

Reverse transcription was performed with 1 μg of total RNA, with the High-Capacity cDNA reverse transcription kit (Applied Biosystems), and MYC and GAPDH and TATA-box binding Protein (TBP) were amplified by PCR in a Roche real-time thermal cycler, with the Roche Taqman master mix (Roche) with the Hs00153408_m1, Hs02758991_g1 and 4326322E assays on demand (encompassing primers and Taqman probes) purchased from Applied Life Technologies.

### DNA array

For the identification of genes displaying changes in expression after the depletion of FGFR3 in MGH-U3 cells, we transfected the cells for 72 h with FGFR3 siRNA#1, FGFR3 siRNA#2 or SMARTpool: ON-TARGETplus FGFR3 siRNA (Dharmacon, L-0031333-00-0005). For the identification of genes displaying a change in expression after FGFR3 depletion in RT112 cells, we transfected the cells for 40 h with FGFR3 siRNA#3 or FGFR3 siRNA#4. mRNA was extracted and purified with RNeasy Mini kits (Qiagen). Total RNA (200 ng) from control and siRNA-treated MGH-U3 and RT112 cells was analyzed with the Affymetrix human exon 1.0 ST DNA array and the Affymetrix U133 plus 2 DNA array, respectively, as previously described for PPARG-regulated genes (Biton et al, 2014). The microarray data described here are available from GEO (https://www.ncbi.nlm.nih.gov/geo/) under accession number GSE84733. The LIMMA algorithm was used to identify genes differentially expressed between FGFR3 siRNA-treated (two and three different siRNAs were used for RT112 and MGH-U3 cells, respectively) and Lipofectamine-treated cells (three replicates; Ritchie et al, 2015). The P-values were adjusted for multiple testing by Benjamini–Hochberg FDR methods. Genes with a $log_2$ fold-change of at least 0.58, in a positive or negative direction, with a FDR below 5%, were considered to be differentially expressed.

### Human bladder samples

We used protein extracted from 129 human bladder tumors (57 non-muscle-invasive and 72 muscle-invasive tumors) for RPPA analysis (Calderaro et al, 2014). The flash-frozen tumor samples were stored at −80°C immediately after transurethral resection or

cystectomy. All tumor samples contained more than 80% tumor cells, as assessed by the hematoxylin and eosin (H&E) staining of histological sections adjacent to the samples used for transcriptome analyses. All subjects provided informed consent, and the study was approved by the institutional review boards of the Henri Mondor, Foch and Institut Gustave Roussy Hospitals. RNA, DNA, and protein were extracted from the surgical samples by cesium chloride density centrifugation, as previously described (Calderaro *et al*, 2014). *FGFR3* mutations were studied with the SNaPshot technique. The expression of FGFR3-TACC3 and FGFR3-BAIAP2L1 was analyzed by PCR, as previously described (Wu *et al*, 2013).

Lyophilized proteins were solubilized in Laemmli sample buffer and boiled for 10 min. Protein concentrations were determined with the Bio-Rad Bradford Protein Assay Kit (Bio-Rad, France).

### Reverse-phase protein array (RPPA)

Reverse-phase protein array with specific anti-phospho-AKT (S473; Cell Signaling Technology # 4058, used at 1/1,000) and anti-phospho-p38 (T180/Y182; BD Biosciences #612288, used at 1/500) antibodies was performed and analyzed as previously described (Calderaro *et al*, 2014). The specificity of the antibodies used for RPPA for the protein of interest was checked by Western blotting with 18 tumor lysates, before the study. We obtained a Pearson coefficient for the correlation between RPPA and Western blotting of 0.84 for P-AKT (66) and 0.88 for P-p38 (data not shown).

### *In vivo* models

Mouse experiments reported herein were approved by Animal Housing and Experiment Board of the French government.

### *Xenograft models*

Six-week-old female Swiss *nu/nu* mice (Charles River Laboratories) were raised in the animal facilities of Institut Curie, in specific pathogen-free conditions. They were housed and cared for in accordance with the institutional guidelines of the French National Ethics Committee (*Ministère de l'Agriculture et de la Forêt, Direction de la Santé et de la Protection Animale*, Paris, France), under the supervision of authorized investigators. Mice received a subcutaneous injection, into each flank (dorsal region), of $5 \times 10^6$ RT112 or MGH-U3 bladder cancer cells in 100 μl PBS. For each study, with each of the cell lines, mice were randomly separated into two groups when tumors reached a volume of 100 mm$^3$ ($\pm$20). For FGFR3 inhibition studies, the mice were treated daily for 9 days, by oral gavage with PD173074 (25 mg/kg; $n = 4$) in one group and with vehicle (0.05 M acetate buffer) in the other ($n = 4$). The tumors were then removed. Part of the tumor was flash-frozen in liquid nitrogen for protein extraction in Laemmli buffer. For p38 inhibition studies, one group received daily injections of SB203580 (100 μl, 20 μM) into the tumor ($n = 5$), whereas the other group received daily injections of vehicle (PBS; $n = 5$). For JQ1 treatment, mice received a daily intraperitoneal injection of 50 mg/kg JQ1 ($n = 6$) or vehicle (10% DMSO, 90% 10% 2-hydroxypropyl β-cyclodextrin; $n = 6$). For each treatment, the tumor was measured twice weekly with calipers, and its volume in mm$^3$ was calculated with the formula: $\pi/6 \times$ (largest diameter) $\times$ (shortest diameter)$^2$.

### *Patient-derived Tumor Xenograft (PDX) model (F659)*

A patient-derived bladder cancer xenograft model (F659) was established as follow. A fresh specimen was collected from a patient diagnosed with a muscle-invasive bladder carcinoma with two positive perivesical lymph nodes (pT3bN2Mx), in accordance with French regulations concerning patient information and consent and then xenografted subcutaneously in the interscapular space of 5-week-old male Swiss *nu/nu* mice (Charles River Laboratories) and serially passaged into male Swiss *nu/nu* mice (Charles River Laboratories). DNA was isolated from snap-frozen tumor from the patient and from the PDX tumor (at passage 3 in mice), with a classical phenol-chloroform-isoamyl alcohol extraction protocol. FGFR3 mutations were studied by the SNaPshot method, as previously described (van Oers *et al*, 2005), and a FGFR3-S249C heterozygous mutation was detected in both samples.

For treatment with the pan-FGFR inhibitor, BGJ398, PDX (F659) tumor tissue at passage 4 in mice was cut into small pieces (5 mm$^3$) and subcutaneously xenografted into multiple mice in the interscapular region. When tumor sizes reached 100–200 mm$^3$, mice were randomly divided into two groups and treated by daily oral gavage with BGJ398 (30 mg/kg, LC Laboratories) or vehicle (0.05 M acetate buffer). Tumor growth was measured twice weekly with an electronic caliper, and tumor volume was calculated and expressed relative to the initial size of the tumor. Two experiments were conducted as follows: one for a long-term treatment (29 days; $n = 5$ animal per group) in which tumors were monitored for two additional weeks after the end of treatment, and one for a short-term treatment over a period of 4 days ($n = 4$ animal per group). The mice were sacrificed at the end of the experiments. Their tumors were harvested and flash-frozen. RNA was isolated with Trizol, and proteins were recovered by lysis in Laemmli buffer for subsequent RT–qPCR and Western blot analyses, respectively.

### Statistical analysis

Linear models for microarray data (LIMMA) was used to analyze DNA array experiments involving simultaneous comparisons between large numbers of RNA targets (Ritchie *et al*, 2015). All functional experiments were carried out twice or three times, in triplicate. Data are expressed as means $\pm$ SD. Tukey's tests were used for multiple comparisons, and unpaired Student's *t*-tests (two-tailed) or Mann–Whitney *U*-tests were used for other statistical analyses. The control siRNA group, the IgG group, or the vehicle group was used as the reference group, depending on the experiment. The RPPA signals of tumors with and without *FGFR3* mutations were compared in Wilcoxon's rank sum tests. Non-parametric Spearman' rank correlation tests were carried out to evaluate the correlation between levels of *MYC* and *FGFR3* mRNA in human bladder tumors.

### Data availability

Transcriptomic data obtained with Affymetrix U133plus2.0 DNA arrays for our CIT-cohorts of bladder tumors, encompassing 82 NMIBCs and 85 MIBCs, were previously deposited on the publicly available ArrayExpress databases E-MTAB-1803 and E-MTAB-1940, respectively (El Behi *et al*, 2013; Biton *et al*, 2014; Rebouissou *et al*, 2014). RNA-Seq data for an independent cohort of 416 tumors were available from ArrayExpress E-MTAB-4321 (Hedegaard *et al*, 2016).

**The paper explained**

**Problem**

Bladder cancer is the ninth most common cancer worldwide. FGFR3 alterations (mutations or translocations) are among the most frequent genetic events in bladder carcinoma. They lead to constitutive activation of the receptor and to oncogene addiction to FGFR3. Anti-FGFR therapies have recently yielded promising results, but the efficacy of such targeted therapies is currently limited by the emergence of resistance. In this study, we investigated the molecular mechanisms underlying the oncogenic activity of activated FGFR3 in bladder tumors, with a view to identifying new drug targets to improve treatment efficacy and/or limit resistance.

**Results**

We identified MYC as a key master regulator of proliferation activated by aberrantly activated FGFR3 in bladder cancer-derived cell lines. We showed that *FGFR3* is a direct target gene of MYC establishing an FGFR3/MYC positive feedback loop. Consistently, we found that human bladder tumors bearing *FGFR3* mutations had levels of *FGFR3* and *MYC* expression that were positively correlated. Further evidence of relevance to human tumors was provided by the use of a PDX model carrying an *FGFR3* mutation, in which FGFR3 inhibition induced a decrease in the expression of both MYC and FGFR3. We demonstrated that this loop was dependent on the activation of p38 and AKT by FGFR3, regulating *MYC* mRNA levels and protein stability, respectively. We showed that p38 and AKT activity were required for FGFR3-induced cell proliferation. Finally, we demonstrated that JQ1, a BET bromodomain inhibitor, was able to prevent *MYC* and *FGFR3* expression. JQ1 treatment significantly decreased cell viability *in vitro* and tumor outgrowth in a xenograft model.

**Impact**

We have identified a novel FGFR3-MYC positive feedback loop in bladder tumor cell lines harboring aberrantly activated FGFR3, which may be of clinical relevance, because it was also found in a PDX model harboring an *FGFR3* mutation. We also provide the first proof of concept that disrupting this loop with various inhibitors of FGFR3, p38, or AKT or with BET bromodomain inhibitors (JQ1) is of potential therapeutic value. These findings open up new possibilities for the treatment of bladder tumors displaying aberrant FGFR3 activation. The simultaneous inhibition of two targets from the same pathway may increase efficacy and prevent the development of resistance, as reported for the use of BRAF and MEK inhibitors for the treatment of melanoma with *BRAF* mutations.

*FGFR3* mutational status and data for eight normal samples were kindly provided by Dr. Ellen Zwarthoff (Erasmus MC Cancer Institute, the Netherlands) and Dr. Lars Dyrskjøt (Aarhus University Hospital, Denmark). The microarray for MGH-U3 and RT112 cells treated with *FGFR3* siRNA are available from GEO (https://www.ncbi.nlm.nih.gov/geo/) under accession number GSE84733.

**Expanded View** for this article is available online.

## Acknowledgements

We thank David Gentien and Leanne De Koning from the genomics and RPPA platforms, respectively, of Institut Curie. We thank Ellen Zwarthoff (Erasmus MC Cancer Institute, the Netherlands) and Dr. Lars Dyrskjøt (Aarhus University Hospital, Denmark) for providing FGFR3 mutations for their cohort of tumors for which transcriptomic data were publicly available and transcriptomic data for eight normal samples, respectively. This work was supported by a grant from *Ligue Nationale Contre le Cancer* (IBP, FR, MM, HNK, RN, CK, MDG) as an associated team (*Equipe labellisée*), the "*Carte d'Identité des Tumeurs*" program initiated, developed, and funded by *Ligue Nationale Contre le Cancer*, the "LIONS" project funded by INSERM/ ITMO Cancer and the "Tumult" project funded by INCa. HNK and VSQ were supported by a fellowship from *Ligue Nationale Contre le Cancer*.

## Author contributions

MM, FD, HN-K, CP, FR, and IB-P designed the study. MM, FD, HN-K, AM-V, CB, MS, IH VS-Q, CK, MD-G, and IB-P performed experiments and analyzed data. FD, EC, and RN carried out bioinformatics analyses. CB, HL, and TM established the PDX model. IB-P supervised the study. MM, FD, HN-K, FR, and IB-P wrote the manuscript. All authors made comments on the manuscript.

## Conflict of interest

The authors declare that they have no conflict of interest.

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
