## [Review Process File · EMBO Molecular Medicine]

An FGFR3/MYC positive feedback loop provides new opportunities for targeted therapies in bladder cancers

Mélanie Mahe, Florent Dufour, H  l  ne Neyret-Kahn, Aura Moreno-Vega, Claire Beraud, Mingjun Shi, Imene Hamaidi, Virginia Sanchez-Quiles, Clementine Krucker, Marion Dorland-Galliot, Elodie Chapeaublanc, Remy Nicolle, Herv   Lang, Celio Pouponnot, Thierry Massfelder, Fran  ois Radvanyi, Isabelle Bernard-Pierrot

Review timeline:

Submission date:	16 June 2017
Editorial Decision:	21 June 2017
Appealed	29 June 2017
Editorial Decision:	06 September 2017
Revision received:	20 December 2017
Editorial Decision:	16 January 2018
Revision received:	19 January 2018
Accepted:	23 January 2018

Editors: Roberto Buccione and C  line Carret

Transaction Report:

1st Editorial Decision

21 June 2017

Thank you for the submission of your manuscript " An FGFR3/MYC positive feedback loop provides new opportunities for targeted therapies in bladder cancers" and apologies for the delay in replying due to increased submissions and concurrent editor travel.

I have now had the opportunity to carefully read your paper and the related literature and I have also discussed it with my colleagues. I am afraid that we concluded that the manuscript is not well suited for publication in EMBO Molecular Medicine and have therefore decided not to proceed with peer review.

Although we acknowledge its potential interest, we agreed that the manuscript is better suited to a cancer biology-dedicated title. In fact, due also to the available knowledge including on the use of FGFR, AKT and BET inhibitors in bladder cancer in vitro and in vivo, we are not persuaded that your manuscript provides the striking conceptual advance and novel clinical and/or translational implications we would like to see for in an EMBO Molecular Medicine article.

I am sorry that I could not bring better news and again, I am sorry for the belated reply.

Appeal (following personal meeting)

29 June 2017

Thank you again for your time and kind consideration regarding our discussion about the manuscript relative to the identification of a FGFR3/MYC positive feedback loop in bladder cancer.

As you proposed, please find attached the external reviews of the manuscript realized for Journal of Clinical Investigation and the correspondence I had with the editor following the rejection of the paper (not shown here).

We decided not to transfer the manuscript to JCI insight but rather improved the manuscript taking into consideration most of the major concerns of the reviewers and to submit it to EMBO Molecular Medicine. In particular, we have added one figure showing the overexpression of MYC in human bladder tumors harboring FGFR3 mutation and the positive correlation between MYC and FGFR3 expression in these tumors, these data support the existence of the FGFR3/MYC loop identified in cell lines in tumors. We have also assessed the expression of other FGFRs in the cell lines used for this study since the inhibitor used is a pan-FGFR inhibitor.

I hope that you will be able to move forward with our work at EMBO mol med.

I look forward to hearing from you.

(Appeal request accepted, manuscript sent out for review)

2nd Editorial Decision

6 September 2017

Thank you for the submission of your manuscript to EMBO Molecular Medicine.

I sincerely apologise for the very unusual delay in getting back to you on your manuscript. In fact, we experienced significant difficulties in securing expert and willing reviewers, and then obtaining their evaluations in a timely fashion, mostly due to the overlapping holiday season. Furthermore, additional internal discussion was required to reach a final decision.

You will see that in aggregate, the reviewers' evaluations paint a mostly positive picture, although #1 and #3 are more reserved. I will not go into much detail but I would like to briefly mention their main concerns: 1) limitations of the study due to the use of established cell lines and the lack advanced in vivo modelling; 2) excessive reliance on correlation rather than causal analysis and 3) lack of a drug combination study.

As you might imagine, the first point indeed gave me pause when deciding whether to send the manuscript out for peer-review and I get the impression, notwithstanding the reviewers' apparently non-negative stance, that this indeed remains a weak point.

This issue and the others were discussed during our cross-commenting exercise and as mentioned above, internally. The consensus that emerged was that although the inclusion of more advanced models (e.g. PDXs) would be valuable, this would possibly require a major undertaking that would not be fully justified.

In conclusion, while publication of the paper cannot be considered at this stage, we are willing to consider a substantially revised manuscript, addressing the reviewers' concerns including with further experimentation where required. I will not, however, be asking you to provide additional data using more advanced in vivo models, although I would encourage you to include such data, if available. It remains important nevertheless, that you discuss the limitations of your study in this respect.

I look forward to receiving your revised manuscript in due time.

***** Reviewer's comments *****

Referee #1 (Comments on Novelty/Model System for Author):

The work is very well performed. The novelty is medium-high; some of the novelty is eclipsed by a recent paper highlighting the role of MYC in the biological effects of FGFRs (Liu et al), cited in the text. However, there are other aspects of the work that are clearly novel. Medical impact is potentially high, mainly because of the potential for the combination of FGFR inhibitors with BET inhibitors (not so novel...). Finally, all the work is performed using cell lines (no primary tumor cultures, no PDXs, no organoids, no genetically modified mice) which somewhat limits the scope of the findings.

Referee #1 (Remarks for Author):

In this paper, Mahe reports a regulatory loop involving mutant constitutively active FGFR3 and the MYC oncogene. FGFR3 activation leads to increased MYC mRNA levels and protein stabilization. In turn, MYC activates FGFR3 expression by binding to regulatory motifs in the enhancer and promoter. The authors also reveal a prominent role of p38 in this process. Importantly, MYC activity in these tumors appears to sensitize them to the effects of the BET protein inhibitor JQ1, as shown both *in vitro* and *in vivo* using a bladder cancer cell line grown as xenograft. Overall, this is a valuable contribution with well performed experiments and novel data that may guide the development of therapeutic strategies in bladder cancer. One is surprised that MYC appears to be in such regulatory loop considering that, in general, MYC-driven tumors are aggressive whereas FGFR3 mutations appear to be much more frequent in the non-muscle invasive bladder cancers. This conundrum would benefit from some discussion.

Major comments

1. The work is performed with established cell lines that may not be completely representative of the original tumors. This limitation may account for some of the effects and, at least, it should be noted in the discussion.
2. Is the binding of MYC to the FGFR3 regulatory regions bladder cancer cell-specific? Have the authors analyzed the many ChIP-Seq datasets available for MYC on the status of FGFR3 chromatin marks in tumors where FGFR3 is not a major player but MYC is? This would provide interesting information regarding the tissue-specific regulation of FGFR3 by MYC.
3. The data on the role of p38 is rather intriguing since it is generally thought that p38 is growth-suppressive. Can the authors speculate, if not provide data, about which context specificity may determine the output of p38 activation?
4. The analysis of the *in vivo* growth was performed only in one cell line, a second line showing consistent results would be important for confirmation. Furthermore, analyzing the activation status of the pathways in the murine tumors would be important.
5. The authors should discuss the work carried out in clinical trials using FGFR inhibitors in bladder cancer.

Referee #2 (Remarks for Author):

The authors have explored some of the downstream signalling cascades that operate in bladder cancer cells expressing commonly mutated forms of FGFR3. The authors show that MYC regulates FGFR3, and that a positive feedback loop exists in which FGFR3 signalling promotes MYC activation and that in turn increases the levels of FGFR3. Experiments with tumour-bearing nude mice demonstrate that drugs predicted to inhibit these pathways [SB203580 (p38) or JQ1 (BET bromodomain)] slow tumour growth.

The work is nicely done and well presented. Little to criticize. I suppose an obvious question is why not test the effects of both drugs at the same time? Toxicity?

Referee #3 (Remarks for Author):

Mahe et al describes important work on FGFR3/MYC regulation in bladder cancer and demonstrates possible therapeutic opportunities in cell lines and in xenograft models. Robust experiments are carried out that focus on transcription factor regulation upon FGFR3 depletion, identification of a positive feed-back loop where FGFR3 is transcribed by MYC, and involvement of p38 and AKT in the regulation.

The findings are novel and highly interesting as FGFR3 mutations are frequently observed in bladder tumors, and novel targeted treatment of the activated pathway may have large clinical impact.

Major points:

1. Most conclusions in this manuscript are based on work carried out in two bladder cancer cell lines with aberrantly activated FGFR3. This is a limitation to the study, and should be discussed.
2. The authors see a correlation (but still huge variation) between MYC expression and FGFR3

expression in FGFR3 mutated tumors using microarray experiments - but not in wt FGFR3 tumors. This analysis could be made stronger by e.g. using the TCGA bladder dataset. Still, this is just a correlation of expression levels and could be caused by several factors (e.g. mRNA stability).

3. Have the authors tried to use a wt cell line (as negative control) in the experiments to determine if the FGFR3/MYC positive feedback loop is dependent on the activated FGFR3 protein? This may underline the importance of the FGFR3 activated feedback loop.
4. Please add more information about the origin of the two cell line used.

Minor points:

1. Line 6 page 4. I would not use the word "rarely" regarding progression (5-40% rage).
2. Reference to IO in introduction - newer articles exist that may be more appropriate to reference. E.g. Bellmunt et al NEJM 2017.
3. Page 6 last line: please explain "CIT" - use another reference

1st Revision - authors' response

6 September 2017

Referee #1

Overall, this is a valuable contribution with well performed experiments and novel data that may guide the development of therapeutic strategies in bladder cancer. One is surprised that MYC appears to be in such regulatory loop considering that, in general, MYC-driven tumors are aggressive whereas FGFR3 mutations appear to be much more frequent in the non-muscle invasive bladder cancers. This conundrum would benefit from some discussion.

This point is now discussed (p20-21).

Major comments:

1. The work is performed with established cell lines that may not be completely representative of the original tumors. This limitation may account for some of the effects and, at least, it should be noted in the discussion.

We agree with the reviewer concerning this limitation of the study. To overcome this limitation, we aimed at validating our results in PDX models which most closely resembles human tumors. Thanks to collaboration with Thierry Massfelder and Hervé Lang (University of Strasbourg), we obtained one PDX model bearing an FGFR3-mutation and we have now added results for this PDX model that confirm our conclusion regarding the identification of the FGFR3/MYC loop in bladder tumors. We show that the PDX were sensitive to an anti-FGFR treatment, BGJ398, which slowed tumor growth (Supplementary Figures 2 and 6B) (p9) and decreased the levels of the MYC and FGFR3 proteins (Figure 2F) (p12). Thus, the results obtained with FGFR3 inhibitors, for PDX tumors and cell line xenografts were similar, increasing our confidence in the relevance of our results to human tumors.

2. Is the binding of MYC to the FGFR3 regulatory regions bladder cancer cell-specific? have the authors analyzed the many ChIP-Seq datasets available for MYC on the status of FGFR3 chromatin marks in tumors where FGFR3 is not a major player but MYC is? this would provide interesting information regarding the tissue-specific regulation of FGFR3 by MYC.

We have now analyzed 118 ChIP-Seq datasets for MYC in different tissues, but the roles of MYC and FGFR3 in these tissues/cell lines are mostly unknown. Binding was observed in two known FGFR3-dependent cell lines, MCF7 and HepG2, in some blood-derived cell lines and in one lung cancer-derived cell line. The regulation of FGFR3 by MYC seemed so to be quite specific to bladder cancer. In line with this observation, MYC activation did not seem to be sufficient to induce FGFR3 regulation. Indeed, MYC ChIP-Seq data acquired for two inducible models of MYC overexpression/activation (LNCaP and U2OS cells) showed no MYC enrichment on the FGFR3 enhancers or promoter after MYC activation. These data have been added to the manuscript ((p11) and Supplementary Figure 5).

3. The data on the role of p38 is rather intriguing since it is generally thought that p38 is growth-suppressive. Can the authors speculate, if not provide data, about which context specificity may determine the output of p38 activation?

We agree that our results are intriguing since p38 was initially reported to display tumor suppressor properties, but numbers of evidence support now the existence of a dual role of p38 in tumor progression. Some other studies that have also highlighted such a pro-proliferative role in other cancer types are cited in the discussion section (p22). However, the contexts in which p38 mediates growth-suppressing or pro-proliferative or pro-migratory signaling, and the factors mediating these different effects, remain unclear. This point is also discussed in the text (p22).

4. The analysis of the in vivo growth was performed only in one cell line, a second line showing consistent results would be important for confirmation. Furthermore, analyzing the activation status of the pathways in the murine tumors would be important.

We previously analyzed the effects of p38 and BET inhibitors only for the *in vivo* growth of MGH-U3 xenografts. We have now also studied the effects of both inhibitors on the *in vivo* growth of RT112 xenografts. The growth inhibition induced by these two inhibitors was similar in the two cell lines, rendering our results more reliable. These data have been added on p15 and Figure 4E for P38 inhibitor and p16 and Figure 5D for JQ1.

Concerning the *in vivo* studies, we showed in PDX-F659 tumors as well as in MGH-U3 and RT112 xenografts that FGFR3 inhibition decreased the MYC and FGFR3 levels. We also observed a phosphorylation of p38 and AKT in these tumors. However, we were unable to demonstrate clearly *in vivo* that the phosphorylation of these two proteins was due to FGFR3 activation. Indeed, we analyzed tumors (PDX and cell xenografts) from mice treated with a pan-FGFR inhibitor and a decrease in p38 phosphorylation was observed in RT112 and PDX tumors but not in MGH-U3 xenografts. An increase in phosphorylation was observed for AKT in some tumors. AKT activation due to EGFR overexpression has clearly been demonstrated to mediate resistance to anti-FGFR therapy (Wang et al., 2017, Eur Urol). Our experimental design may not, therefore, be suitable for addressing this question, due to the early development of resistance/adaptation under treatment. The results are provided as a figure for the reviewer, but have not been integrated into the manuscript, due to their heterogeneous and uninformative nature.

5. The authors should discuss the work carried out in clinical trials using FGFR inhibitors in bladder cancer.

The various studies relating to clinical trials of FGFR inhibitors in bladder cancer have now been discussed (p5 and p23).

Referee #2

The work is nicely done and well presented. Little to criticize. I suppose an obvious question is why not test the effects of both drugs at the same time? Toxicity?

We have now tested the effects of both drugs used at the same time on the viability of RT112 and MGH-U3 cells. The *in vitro* study made it possible to use ranges of doses for both molecules and, therefore, to determine whether any additive or synergistic effects could be observed. We found that inhibition was significantly stronger, resulting in lower viability, when the two drugs were used together than when they were used separately. Mathematical analysis of our results with Loewe methods highlighted an additive effect of the two drugs and even suggested that a synergistic effect might occur for some concentrations of the inhibitors. These results are now described in the manuscript, p17, Figure 5E and Supplementary Figure 9.

It was not obvious how best to transpose the drug doses used *in vitro* so as to obtain these additive or synergistic effects *in vivo* and we had no data concerning the toxicity of such a combination, so we did not take ours *in vivo* studies any further at this point. However, such work is currently underway for our PDX model.

Referee #3

Major points:

1. Most conclusions in this manuscript are based on work carried out in two bladder cancer cell lines with aberrantly activated FGFR3. This is a limitation to the study, and should be discussed.

We agree that our work was mostly based on bladder cancer cell lines (2 to 4, depending on the experiments). The relevance of our conclusions to human tumors was suggested only by the correlation data. Thanks to collaboration with Thierry Massfelder and Hervé Lang (University of Strasbourg), we obtained one PDX model bearing an FGFR3 mutation. We have now added results for this PDX model that confirm our conclusion regarding the identification of the FGFR3/MYC loop in bladder tumors and, therefore, the clinical implications of inhibiting this loop. We show that the PDX were sensitive to an anti-FGFR treatment, BGJ398, which slowed tumor growth (Supplementary Figure 2) and decreased the levels of the MYC and FGFR3 proteins (Figure 2F). Thus, the results obtained with FGFR3 inhibitors, for PDX and cell line xenografts were similar, increasing our confidence in the relevance of our results to human tumors.

2. The authors see a correlation (but still huge variation) between MYC expression and FGFR3 expression in FGFR3 mutated tumors using microarray experiments - but not in wt FGFR3 tumors. This analysis could be made stronger by e.g. using the TCGA bladder dataset. Still, this is just a correlation of expression levels and could be caused by several factors (e.g. mRNA stability).

We strengthened our data showing that MYC is more strongly expressed in tumors with FGFR3 alterations than in normal samples and that FGFR3 and MYC mRNA levels are positively correlated, by analyzing another publically available bladder tumor dataset. We did not use the TCGA dataset since it includes peritumoral tissues as normal samples presenting surprisingly an over-expression of MYC as compared to tumors. Furthermore it encompasses only MIBC presenting a quite low rate of FGFR3 mutations. Instead, we used a dataset enriched in NMIBCs (402/416), in which FGFR3 is frequently mutated, and eight normal urothelium samples (Hedegaard et al., cancer cell, 2016). The results obtained with this dataset confirmed those obtained with our dataset. These new correlation data have been added to the revised manuscript as Supplementary Figure 1 and p8. These data are, indeed, just correlation data suggesting that FGFR3 might regulate MYC expression, and they provide no information about the functional relationship between FGFR3 activity and MYC expression. However, our new data for a PDX model bearing an *FGFR3* mutation reveal a decrease in MYC mRNA levels following FGFR3 inhibition, providing further support for this hypothesis (Figure 1E) (p8).

3. Have the authors tried to use a wt cell line (as negative control) in the experiments to determine if the FGFR3/MYC positive feedback loop is dependent on the activated FGFR3 protein? This may underline the importance of the FGFR3 activated feedback loop.

We have now assessed the impact of FGFR3 inhibition on the expression of MYC and FGFR3 in a bladder cancer cell line expressing wild-type FGFR3, the UM-UC-5 cell line. Our results indicate that the FGFR3/MYC positive feedback loop is, indeed, dependent on the aberrantly activated FGFR3 protein. These results have been added to the manuscript (p12) and figure 2D.

4. Please add more information about the origin of the two cell line used.

More information about the origin of the cell lines used has now been added to the materials and method section (p25)

Minor points:

1. Line 6 page 4. I would not use the word "rarely" regarding progression(5-40% rage).

The word “rarely” has been replaced by the word “sometimes”.

2. Reference to IO in introduction - newer articles exist that may be more appropriate to reference. E.g. Bellmunt et al NEJM 2017.

Recent references relating to immunotherapy have been added, including Bellmunt et al. NEJM 2017.

3. Page 6 last line: please explain "CIT" - use another reference

CIT stand for "Carte d'Identité des Tumeurs" (Tumor identity card). The description of the publicly available data from the CIT program has been modified and moved to the materials and methods section (p32). A reference is now supplied for each dataset.

Analysis of p38 and AKT phosphorylation in RT112 and MGH-U3-derived xenograft tumors and PDX tumors bearing FGFR3-S249C mutation from mice treated with a pan FGFR inhibitor.

- (A) RT112 and MGH-U3-derived xenograft tumors from mice treated for nine days with vehicle or PD173074 (25 mg/kg/day, same tumors than the one showed in Figure 2E) were lysed and lysates were immunoblotted with antibodies against FGFR3, MYC, AKT, phospho-AKT (P-AKT Ser413), phospho-p38 (P-p38 Thr180/Tyr182), and p38.
- (B) PDX tumors bearing FGFR3-S249C mutation from mice treated for four days with vehicle or BGJ398 (30 mg/kg/day) were lysed (same tumors than the one showed in Figure 2F) and the protein levels of FGFR3, MYC, AKT, phospho-AKT (P-AKT Ser413), phospho-p38 (P-p38 Thr180/Tyr182), and p38 were assessed by western blotting. Actin was used as a loading control.

Thank you for the submission of your revised manuscript to EMBO Molecular Medicine. We have now received the enclosed report from the referee who was asked to re-assess it. As you will see the reviewer is now supportive and I am pleased to inform you that we will be able to accept your manuscript pending a few final amendments.

***** Reviewer's comments *****

Referee #3 (Comments on Novelty/Model System for Author):

The authors have addressed my previous concerns adequately, and I have no further questions to this.

Referee #3 (Remarks for Author):

The authors have addressed my previous concerns adequately, and I have no further questions to this.

Corresponding Author Name: Isabelle Bernard-Pierrot

Manuscript Number: EMM-2017-08163-V2-Q